# SCALING VALUE ITERATION NETWORKS TO 5000 LAYERS FOR EXTREME LONG-TERM PLANNING

## ABSTRACT

The Value Iteration Network (VIN) is an end-to-end differentiable architecture that performs value iteration on a latent Markov Decision Process (MDP) for planning in reinforcement learning (RL). However, VINs struggle to scale to long-term and large-scale planning tasks, such as navigating a $100 \times 100$ maze—a task that typically requires thousands of planning steps to solve. We observe that this deficiency is due to two issues: the representation capacity of the latent MDP and the planning module's depth. We address these by augmenting the latent MDP with a dynamic transition kernel, dramatically improving its representational capacity, and, to mitigate the vanishing gradient problem, introduce an "adaptive highway loss" that constructs skip connections to improve gradient flow. We evaluate our method on 2D maze navigation environments, the ViZDoom 3D navigation benchmark, and the real-world Lunar rover navigation task. We find that our new method, named *Dynamic Transition VIN (DT-VIN)*, scales to 5000 layers and solves challenging versions of the above tasks. Altogether, we believe that DT-VIN represents a concrete step forward in performing long-term large-scale planning in RL environments.

## 1 INTRODUCTION

Planning is the problem of finding a sequence of actions that achieve a specific pre-defined goal. As the aim of both some older algorithms (e.g., Dyna (Sutton, 1991), A* (Hart et al., 1968), and others (Schmidhuber, 1990a;b)) and many recent ones (e.g., the Predictron (Silver et al., 2017), the Dreamer family of algorithms (Hafner et al., 2020; 2021; 2023), SoRB (Eysenbach et al., 2019), SA-CADRL (Chen et al., 2017), and the LLM-planner (Song et al., 2023)), effective planning is a long-standing and important challenge in artificial intelligence (AI).

Traditional search-based planning algorithms like A* require an accurate environmental model. Thus, these algorithms are limited in their effectiveness when faced with unknown Markov decision processes. In such scenarios, a policy can be learned either through imitation learning (IL), which leverages expert demonstrations, or through trial and error with reinforcement learning (RL). Within RL and IL, the Value Iteration Network (VIN) (Tamar et al., 2016) stands out as quite unique due to its distinctive architecture that integrates a differentiable latent "planning module" into the deep neural network, rather than maintaining an explicit learned environment model like Dreamer (Hafner et al., 2020) or MuZero (Schrittwieser et al., 2020). This integrated architecture allows VINs to be trained end-to-end, meaning that both the state representation components and the planning components are implicitly trained jointly. Additionally, the integrated planning capability of VINs still enables them to effectively generalize to related but unseen tasks. VINs have been shown to perform well in some small-scale short-term planning situations, like path planning (Pflueger et al., 2019; Jin et al., 2021), autonomous navigation (Wöhlke et al., 2021), and complex decision-making in dynamic environments (Li et al., 2021). However, they still struggle to solve larger-scale and longer-term planning problems. We refer to *large-scale planning tasks* as those with high-dimensional observation space (e.g., the maze size), and *long-term planning tasks* as those necessitating extended planning horizons to achieve the goal. For example, in a $100 \times 100$ maze navigation task, the success rate of VINs in reaching the goal drops to well below 40% (see Figure 1(b)). Even in smaller $35 \times 35$ mazes, the success rate of VINs drops to 0% when the required planning steps exceed 60 (see Figure 1(c)).

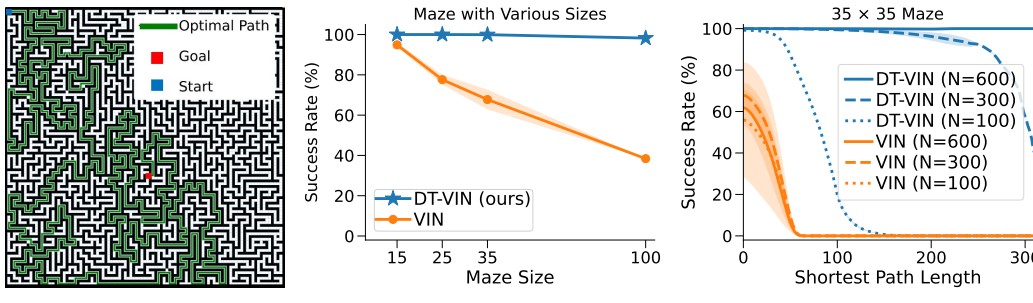

(a) $100 \times 100$ Maze Navigation  (b) Performance for Different Domain Scales  (c) Performance for Different Planning Steps

Figure 1: (a) shows an example of $100 \times 100$ maze navigation task, where the green line shows the optimal path from the start position (blue) to the goal position (red). See Figure 8 in Appendix B for more examples of mazes with other sizes. (b) shows the success rate of VIN and DT-VIN on the maze navigation tasks as a function of maze size. The reported results are computed in expectation over different shortest path lengths for each maze size. (c) shows the success rate of VIN (Tamar et al., 2016) and our DT-VIN as a function of planning steps on the $35 \times 35$ maze benchmark.

Our work identifies that the principal deficiency causing this is the mismatch between the complexity of planning and the comparatively weak representational capacity of the relatively shallow networks that it uses. And while there has been moderate success in learning more complicated networks (e.g., GPPN (Lee et al., 2018) and Highway VINs (Wang et al., 2024a)), until now, VINs of a scale capable of long-term or large-scale planning have not been computationally tractable due to persistent issues with vanishing and exploding gradients—a fundamental problem of deep learning (Hochreiter, 1991).

In this work, we aim to surgically correct deficiencies in VIN-based architectures to enable large-scale long-term planning. Specifically, we first identify the limitations of the latent MDP in the planning module of VIN and propose a dynamic transition kernel to dramatically increase the representational capacity of the network. We then build on existing work that identifies the connection between network depth and long-term planning (Wang et al., 2024a) and propose an "adaptive highway loss" that selectively constructs skip connections to the final loss according to the actual number of planning steps. This approach helps mitigate the vanishing gradient problem and enables the training of very deep networks. With these changes, we find that our new *Dynamic Transition Value Iteration Network* (*DT-VIN*), is able to be trained with 5000 layers and scale to 1800 planning steps in a $100 \times 100$ maze navigation task (compared to the original VIN, which only scaled to 120 planning steps in a $25 \times 25$ maze). We apply our method to top-down image-based maze navigation tasks, the first-person image-based ViZDoom benchmark (Wydmuch et al., 2019), and real-world Lunar rover navigation tasks (Berlin, 2018). We find that DT-VINs can solve both despite these problems requiring hundreds to thousands of planning steps. Together, these demonstrate the practical utility of our method on vision-based tasks that previous methods are simply unable to solve. This also serves to highlight the potential of our method to scale to increasingly complex planning tasks alongside the increasing availability of computing power.

## 2 Preliminaries

**Reinforcement Learning (RL) and Imitation Learning (IL).** The most common formalism used for RL is that of the Markov Decision Process (MDP) (Bellman, 1957). We consider an MDP—as per Puterman (2014)—to be the 6-tuple $(\mathcal{S}, \mathcal{A}, \mathcal{T}, \mathcal{R}, \gamma, \mu)$, where $\mathcal{S}$ is a countable state space, $\mathcal{A}$ is a finite action space, $\mathcal{T}(s'|s, a)$ represents the probability of transitioning to state $s' \in \mathcal{S}$ when being in state $s \in \mathcal{S}$ and taking action $a \in \mathcal{A}$, $\mathcal{R}(s, a, s')$ is the scalar reward function, $\gamma \in [0, 1)$ is a discount factor, and $\mu$ is a distribution over initial states. The behaviour of an artificial agent in an MDP is defined by its policy $\pi(a|s)$, which specifies the probability of taking action $a$ in state $s$. The state value function $V^\pi(s)$ is the expected discounted sum of rewards from state $s$ and following policy $\pi$, i.e., $V^\pi(s) \triangleq \mathbb{E}\left[\sum_{t=0}^{\infty} \gamma^t \mathcal{R}(s_t, a_t, s_{t+1})|s_0 = s; \pi\right]$. The goal of RL is usually to find an optimal policy $\pi^*$ that achieves the highest expected discounted sum of rewards. The value function

of an optimal policy is denoted by $V^*(s) = \max_\pi V^\pi(s)$, and satisfies $V^{\pi^*}(s) = V^*(s) \forall s$. The Value Iteration (VI) algorithm iteratively applies the following update to all states to obtain the optimal value function: $V^{(n+1)}(s) = \max_a \sum_{s'} \mathcal{T}(s'|s,a) \left[ \mathcal{R}(s,a,s') + \gamma V^{(n)}(s') \right]$, where $n$ is the iteration number. In situations where designing a comprehensive reward function is difficult, imitation learning (IL) offers a practical alternative. IL techniques enable an agent to learn behaviors by observing demonstrations from human or algorithmic experts. Approaches such as Behavioral Cloning directly mimic the actions of an expert in similar states, whereas Inverse Reinforcement Learning (IRL) (Ng et al., 2000) involves inferring the underlying reward function based on the expert's behavior, thereby enabling the agent to optimize its own policy (Schaal, 1996).

**Value Iteration Networks (VINs).** VIN is an end-to-end differentiable neural network architecture for planning which demonstrates strong generalization to unseen domains through the incorporation of an explicit planning module (Tamar et al., 2016). The main idea of VIN is to map observations into a latent MDP $\overline{\mathcal{M}}$ and then use the embedded planning module to perform value iteration (VI) on this latent MDP. Below, we use $\overline{\cdot}$ to denote all the terms associated with the latent MDP $\overline{\mathcal{M}}$.

For each decision, VIN first maps an observation $x$, e.g., an image of a maze and the current position of the agent, to $\overline{\mathcal{M}}$. $\overline{\mathcal{M}}$ is described by the latent state space $\overline{\mathcal{S}} = \{(i,j)\}_{i,j \in [m]}$; a fixed discrete latent action space $\overline{\mathcal{A}}$; a latent reward matrix $\overline{\mathsf{R}} = f^{\overline{\mathsf{R}}}(x) \in \mathbb{R}^{m \times m}$, where $f^{\overline{\mathsf{R}}}$ is a learnable NN called a *reward mapping module*; and a latent transition matrix (or kernel) $\overline{\mathsf{T}}^{\text{inv}} \in \mathbb{R}^{|\overline{\mathcal{A}}| \times F \times F}$ with $F$ representing the dimension of the kernel. *The latent transition matrix is a parameter matrix that is invariant for each latent state $(i,j)$, independent of the observation $x$*, and not restricted to satisfy the probabilistic property, i.e., its elements are not required to represent probabilities or sum to one. Next, VIN conducts VI on the latent MDP $\overline{\mathcal{M}}$ to approximate the latent optimal value function $\overline{V}^*$. To ensure the differentiability of the VI computation, a differentiable VI module is proposed. This module simulates VI computation using differentiable CNN operations, i.e., convolutional and max-pooling operations: $\overline{V}_{i,j}^{(n)} = \max_{\overline{a}} \sum_{i',j'} \overline{\mathsf{T}}_{\overline{a},i',j'}^{\text{inv}} \left( \overline{\mathsf{R}}_{i-i',j-j'} + \overline{V}_{i-i',j-j'}^{(n-1)} \right)$, $i,j \in [m]$. This equation sums over a matrix patch centered around position $(i,j)$.

After the above, by stacking the VI module for $N$ layers, the latent value function is then fed to a policy mapping module by $f^\pi$ to represent a policy that is applicable to the actual MDP $\mathcal{M}$. Here, $f^\pi \left( \overline{V}^{(n)}(x), a \right)$ represents the probability of taking action $a$ given observation $x$. Finally, the model can be trained by standard RL and IL algorithms with the following general loss: $\mathcal{L}(\theta) = \frac{1}{|\mathcal{D}|} \sum_{(x,y) \in \mathcal{D}} \ell \left( f^\pi \left( \overline{V}^{(N)}(x), \cdot \right), y \right)$, where $\mathcal{D} = \{(x,y)\}$ is the training data, $x$ is the observation, $y$ is the label, and $\ell$ is the sample-wise loss function. The specific meaning of these items varies depending on the task. For example, in imitation learning, where the expert data is provided, the label $y$ is the expert action and $\ell$ is the cross-entropy loss, i.e., $\ell \left( f^\pi \left( \overline{V}^{(N)}(x) \right), y \right) = -\sum_{a \in \mathcal{A}} \mathbb{1}_{\{a=y\}} \log f^\pi \left( \overline{V}^{(n)}(x), a \right)$, where $\mathbb{1}$ is the indicator function.

# 3 METHOD

In this section, we discuss how to train scalable VINs for long-term large-scale planning tasks. Our method addresses the two key issues with VIN that are identified as hampering its scalability: the capacity of the latent MDP representation and the depth of the planning module.

## 3.1 INCREASING THE REPRESENTATION CAPACITY OF THE LATENT MDP

**Motivation.** VIN utilizes the computational similarities between VI and CNNs to directly implement VI through a CNN-based VI module, as described in Section 2. However, there is a discrepancy between the CNN-based VI module and the general VI computation process.

*CNN-based VIN uses an* invariant *latent transition kernel* $\overline{\mathsf{T}}^{\text{inv}} \in \mathbb{R}^{|\overline{\mathcal{A}}| \times F \times F}$ *as a learnable parameter, which is the same for each latent state $\overline{s} = (i,j)$ and independent of the current observation, e.g., the map of the maze.* This severely limits the representational capacity of the latent MDP which, to be effective, should model what will in practice be the complex and state-dependent transition

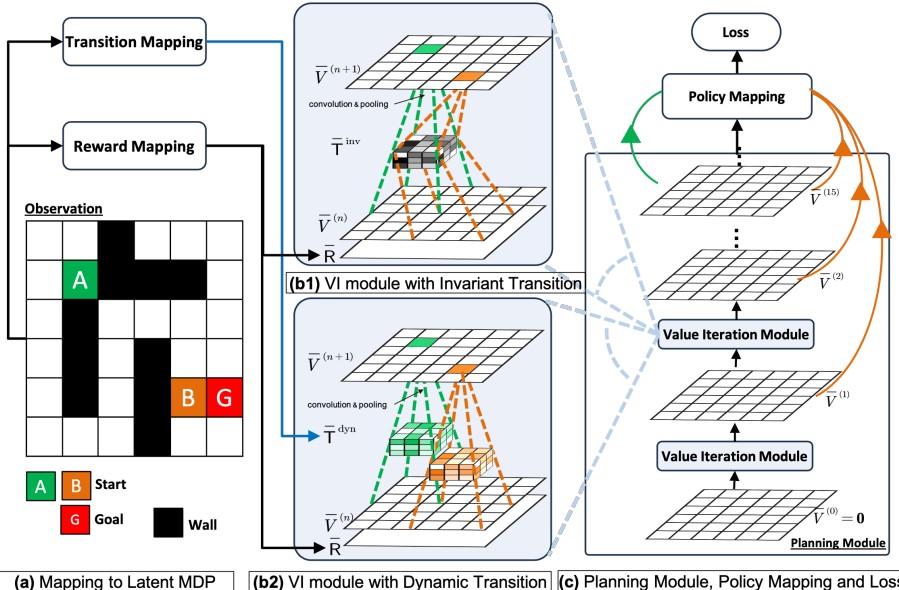

Figure 2: The architecture of VIN and DT-VIN in the maze navigation task. (a) shows the observation of the maze, which is mapped to the latent reward/transition matrix of the latent MDP through the reward/transition mapping module. (c) shows the "planning module", the policy mapping module and the loss. The "planning module" contains numerous stacked Value Iteration (VI) modules. The green and orange connections show an example of adaptive highway loss for planning tasks starting from A and B, respectively. (b1) shows the VI module of the original VIN with invariant transition $\overline{\mathsf{T}}^{\mathrm{inv}} \in \mathbb{R}^{|\mathcal{A}| \times F \times F}$. (b2) shows the VI module of DT-VIN with dynamic transition kernel $\overline{\mathsf{T}}^{\mathrm{dyn}} \in \mathbb{R}^{m \times m \times |\mathcal{A}| \times F \times F}$.

function of the actual MDP. For example, in the maze navigation problem shown in Figure 1(a), the transition probabilities are quite different if the adjacent cell is a wall versus an empty cell. Additionally—as the latent transition kernel of VIN is independent of the real observation—VIN is unable to exploit any information in the observations to simultaneously model the different transition dynamics of different environments. In the maze example, this means that it will greatly struggle because the model is employed to plan on completely different mazes. Altogether, this lack of representation capacity does not affect VIN's performance in small-scale, short-term planning tasks (as were tested on in the original work) where the state space is limited and only a few steps are needed to reach the goal. However, we found it to be a major barrier to VIN's effectiveness in large-scale, long-term planning tasks. As we have shown in Figure 1(c), VIN fails on large-scale $100 \times 100$ maze navigation tasks and long-term planning tasks requiring more than 60 steps.

**Method.** Due to the above, we aim to increase the representation capacity of VIN's latent MDP. To this end, we propose a new architecture called *Dynamic Transition VINs (DT-VINs)*. Instead of using an invariant latent transition kernel, DT-VINs employ a dynamic latent transition kernel $\overline{\mathsf{T}}^{\mathrm{dyn}} = f^{\overline{\mathsf{T}}}(x) \in \mathbb{R}^{m \times m \times |\overline{\mathcal{A}}| \times F \times F}$. For this dynamic kernel, we adhere the same framework outlined in the original VIN paper, which inputs the observation into a learnable *transition mapping module* $f^{\overline{\mathsf{T}}}$.[1] With a dynamic transition kernel, we condition on the latent state $\overline{s} = (i, j)$, allowing it to vary alongside $\overline{s}$, whereas in classical VIN, the kernels $\overline{\mathsf{T}}^{\mathrm{inv}} \in \mathbb{R}^{\overline{\mathcal{A}} \times F \times F}$ remain invariant across all latent states $\overline{s}$. The augmented *dynamic transition VI module* is computed as follows:

$$\overline{V}_{i,j}^{(n)} = \max_{\overline{a}} \sum_{i',j'} \overline{\mathsf{T}}_{i,j,\overline{a},i',j'}^{\mathrm{dyn}} \left( \overline{\mathsf{R}}_{i-i',j-j'} + \overline{V}_{i-i',j-j'}^{(n-1)} \right). \tag{1}$$

---

[1]Although the original VIN paper proposes a general framework where the latent transition kernel depends on the observation, i.e., $\overline{\mathsf{T}} = f^{\overline{\mathsf{T}}}(x)$, they implement it as an independent parameter.

The transition mapping module $f^{\overline{\mathsf{T}}}$ can be any type of neural network, such as CNNs or fully connected networks. In our Maze Navigation tasks, $f^{\overline{\mathsf{T}}}$ includes only one convolutional layer with a kernel size of $F \times F$, which iteratively maps each local patch of the maze to a $|\overline{\mathcal{A}}| \times F \times F$ latent transition kernel for each latent state. This architecture requires $|\overline{\mathcal{A}}|F^4$ number of parameters, compared to the original VIN's $|\overline{\mathcal{A}}|F^2$. Note that in practice, a small kernel size $F$ of 3 is used and is sufficient to produce strong performance. Thus, this alternative module greatly improves the representation capacity of VIN, but typically does not introduce a significant change in training cost.

## 3.2 Increasing Depth of Planning Module

**Motivation.** Recent work on Highway VIN has demonstrated the relationship between the depth of VIN's planning module and its planning ability (Wang et al., 2024a). A deeper planning module implies more iterations of the value iterations process, which is proved to result in a more accurate estimation of the optimal value function (see Theorem 1.12 (Agarwal et al., 2019)). However, training very deep neural networks is challenging due to the vanishing or exploding gradient problem (Hochreiter, 1991). Highway VINs address this issue by incorporating skip connections within the context of reinforcement learning, showing similarities to existing works for classification tasks (Srivastava et al., 2015a; He et al., 2016). Although Highway VINs can be trained with up to 300 layers, they still fail to achieve perfect scores in larger-scale and longer-term planning tasks and necessitate a more intricate implementation. Here, we present a more simple, easy-to-implement method for training very deep VINs.

**Method.** To facilitate the training of very deep VINs, we also adopt the skip connections structure but implement it differently. Our central insight is that short-term planning tasks generally require fewer iterations of value iteration compared to long-term planning tasks. This is because the information from the goal position propagates to the start position in fewer steps when their distance is short. Therefore, we propose adding additional loss to shallower layers directly when the task requires only a few steps. We achieve this by introducing the following *adaptive highway loss*:

$$\mathcal{L}(\theta) = \frac{1}{K|\mathcal{D}|} \sum_{(x,y,l)\in\mathcal{D}} \sum_{1\leq n\leq N} \mathbb{1}_{\{n\geq l\}} \ell\left(f^\pi\left(\overline{V}^{(n)}(x),\cdot\right), y\right), \tag{2}$$

Here, $K = \sum_{(x,y,l)\in\mathcal{D}} \sum_{1\leq n\leq N} \mathbb{1}_{\{n\geq l\}}$, $\mathbb{1}$ is the indicator function, and $l$ is the length of planning path or trajectory, which can be computed from the training data. For example, in the imitation learning of the maze navigation task, for each maze in the dataset, $l$ is the length of the provided expert path from the start to the goal, and the loss function in Equation (2) can be written as $\mathcal{L}(\theta) = \frac{1}{K|\mathcal{D}|} \sum_{(x,y,l)} \sum_n \mathbb{1}_{\{n\geq l\}} \left(-\sum_a \mathbb{1}_{\{a=y\}} \log f^\pi\left(\overline{V}^{(n)}(x), a\right)\right)$. In RL, where the policies are learned through policy gradient, the loss function can be rewritten as $\mathcal{L}(\theta) = \frac{1}{K|\mathcal{D}|} \sum_{(x,y,R,l)\in\mathcal{D}} \sum_{1\leq n\leq N} \mathbb{1}_{\{n\geq l\}} \left(-R \log f^\pi\left(\overline{V}^{(n)}(x), y\right)\right)$, where $y$ is the excuted action, and $R$ the cumulative future reward. As Equation (2) implies, it constructs skip connections for the hidden layers to improve information flow, similar to existing works such as Highway Nets and Residual Nets (Srivastava et al., 2015a; He et al., 2016). However, we connect the hidden layers directly to the final loss, while existing works typically connect skip connections between the intermediate layers. Note that we construct skip connections for each layer $n \geq l$ rather than at the specific layer $n = l$. This is because it would be beneficial for a relatively deeper VIN with depth $n > l$ to also output the correct action in short-term planning tasks. Additionally, during the execution phase, the actual planning steps are unknown, so only the output of the last layer of the VIN will be used. Note that this additional loss will not alter the inherent structure of the value iteration process and will be removed during the execution phase. To avoid the gradient exploding problem, we enforce a softmax operation on the values of the latent transition kernel for each latent state $\overline{s}$. This gives a statistical semantic meaning to the latent transition kernel. This change is simple but critical to training stability, as will be shown in experimental results in Section 4.1 and Figure 4(d).

## 4    EXPERIMENTS

We perform several experiments to test if our modifications to VIN's planning module allow training very deep DT-VINs for large-scale long-term planning tasks. Following previous work (Tamar et al., 2016; Lee et al., 2018), we focus on the imitation learning scenario, where we leverage expert demonstrations to evaluate planning capabilities. IL offers a more stable and controlled setting by reducing the variability that typically arises from the exploratory processes in RL. In line with previous works (e.g., (Lee et al., 2018)), we assess our planning algorithms on navigation tasks within 2D mazes and 3D ViZDoom (Wydmuch et al., 2019) environments (see Sections 4.1 and 4.2, respectively). Each task includes a start position and a goal position, and the agent navigates the four adjacent cells by moving one step at a time in any of the four cardinal directions. Our experiments look at each method's effectiveness over several versions of the tasks with the different versions having different *shortest path lengths (SPLs)*. The SPLs are precomputed using Dijkstra's algorithm and serve as a good proxy measure for the complexity of the planning task. We say that an agent has succeeded in a task if it generates a path from the start position to the goal position within a predetermined number of steps ($m^2$ in our paper). We further say that the agent has found an optimal path if the corresponding path has a minimal length. We follow GPPN and use these for the *success rate (SR)*, which is the rate at which the algorithm succeeds in the task, and the *optimality rate (OR)*, which is the rate at which the algorithm generates an optimal path. In addition to the above, we also test the generality of the DT-VIN approach to two additional tasks. In the style of a benchmark examined in the original VIN paper, a lunar rover navigation task (see Section 4.3); and, to demonstrate the potential for complex action spaces, a continuous control task (see Appendix C.3).

On the above tasks, we compare our DT-VIN method with several advanced neural networks designed for planning tasks, including the original VIN (Tamar et al., 2016), GPPNs (Lee et al., 2018), and Highway VIN (Wang et al., 2024a). The models are trained through imitation learning using a labelled dataset. We then identify the best-performing model based on its results on a validation dataset and evaluate it on a separate test dataset. Following the methodology from the GPPN paper, we conduct evaluations using three different random seeds for each algorithm. This is sufficient to provide a reliable performance estimate here due to the low standard deviation we observe in the tasks. All figures that show learning curves report the mean and standard deviation on the test set.

### 4.1    2D MAZE NAVIGATION

**Setting.**    In our evaluation, we use 2D maze navigation tasks with sizes $M$ set to 15, 25, 35, and 100. Many of these mazes require hundreds or thousands of planning steps to be solved. To assess the performance of each algorithm, we test various neural network depths $N$. Specifically, for mazes of size $M = 15$ and $M = 25$, we examine depths in $N = 30, 100, 200$. For $M = 35$, we examine depths in $N = 30, 100, 300, 600$. For the largest mazes, $M = 100$, we examine depths of $N = 600, 5000$. For each maze size, we generate a dataset following the methodology in GPPN (Lee et al., 2018). Each sample has a starting position, a visual representation of the $m \times m$ map, and an $m \times m$ matrix indicating the position of the goal. For more details, see Appendix B.1.

**Results and Discussion.**    Figure 3(a) and Table 1 show the success rates (SRs) of our method and the baseline methods, as a function of the SPLs. For each algorithm and environment configuration, we report the performance of the NN with the best depth $N$ across the ranges specified in the previous paragraph (see Figure 9 in Appendix C for other values of $N$). Here, DT-VIN outperforms all the other methods on all the maze navigation tasks under all the various sizes $M$ and SPLs. Notably, on small-scale mazes with size in $M = 15, 25, 35$, DT-VIN achieves approximately 100% SRs on all the tasks. For the most challenging environment with $M = 100$, DT-VIN performs best with the full 5000 layers, and it maintains an SR of approximately 100% on short-term planning tasks with SPL ranging in $[1, 200]$ and an SR of approximately 88% on tasks with SPLs over 1200. Comparatively, VIN performs well on small-scale and short-term planning tasks. However, even on a small-scale maze with size $M = 15$, VIN's SRs drop to 0% when the SPL exceeds 30. Moreover, when the maze size increases to 100, VIN only achieves an SR of less than 40%—even on short-term planning tasks with SPL within $[1, 100]$. GPPN performs well on short-term planning tasks, but it fails to generalize well on long-term planning tasks, which also decreases to an SR of 0% as the SPL increases. Highway VIN performs well across tasks with various SPLs on a small-scale maze with $M = 15, 25$. However, it shows a performance decrease on larger-scale maze tasks with

Table 1: The success rates for each method under tasks with different ranges of shortest path length. For each algorithm, we choose the best result from a range of depths. Specifically, for our DT-VIN, the optimal depth consistently corresponds to the maximum value in the range: 600 for size 35, and 5000 for size 100. For other compared methods, the optimal depth differs depending on the task. In the maze of size 100, the optimal depth for all the baselines is 600. For additional results, see Figure 9 in Appendix C.

| Maze Size | $35 \times 35$ | | | $100 \times 100$ | | |
|---|---|---|---|---|---|---|
| SPL | [1,100] | [100, 200] | [200, 300] | [1,600] | [600, 1200] | [1200, 1800] |
| VIN (Tamar et al., 2016) | $68.41_{\pm 6.25}$ | $0.0_{\pm 0.00}$ | $0.00_{\pm 0.00}$ | $45.05_{\pm 0.04}$ | $0.00_{\pm 0.00}$ | $0.00_{\pm 0.00}$ |
| GPPN (Lee et al., 2018) | $95.71_{\pm 0.33}$ | $0.39_{\pm 0.27}$ | $0.00_{\pm 0.00}$ | $75.72_{\pm 0.64}$ | $0.00_{\pm 0.00}$ | $0.00_{\pm 0.00}$ |
| Highway VIN (Wang et al., 2024a) | $90.67_{\pm 3.92}$ | $65.50_{\pm 5.59}$ | $54.40_{\pm 10.2}$ | $69.12_{\pm 0.02}$ | $0.00_{\pm 0.00}$ | $0.00_{\pm 0.00}$ |
| DT-VIN (ours) | $\mathbf{100.00_{\pm 0.00}}$ | $\mathbf{99.99_{\pm 0.01}}$ | $\mathbf{99.77_{\pm 0.23}}$ | $\mathbf{99.98_{\pm 0.00}}$ | $\mathbf{99.56_{\pm 0.20}}$ | $\mathbf{88.65_{\pm 4.76}}$ |

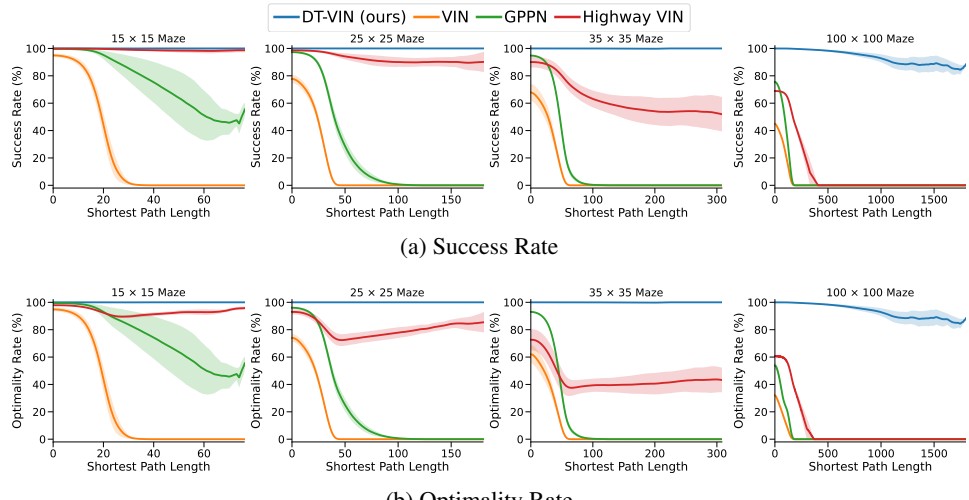

(a) Success Rate

(b) Optimality Rate

Figure 3: SRs and ORs for different algorithms as a function of the shortest path length and the maze size. For each algorithm, we select the best result across various depths. Specifically, for our DT-VIN, the optimal depth consistently corresponds to the maximum value in the range: 200 for mazes of size 15 and 25, 600 for size 35, and 5000 for size 100. For other methods, the optimal depth differs per task. In the maze of size 100, the optimal depth for all the baselines is 600. See Figure 9 and Figure 10 in Appendix C for additional results at other depths.

$M = 35, 100$. Figure 3(b) shows the optimality rates (ORs) of the algorithms, which measure the rate at which the model outputs the optimal path. Our DT-VIN maintains consistent ORs compared to SRs. However, some other methods—especially Highway VIN—exhibit a clear decrease in ORs, indicating that the paths generated by these methods is often sub-optimal.

**Ablation Study.** We perform multiple ablation studies with a $M = 35$ maze and an NN with depth $N = 600$ to assess the impact on DT-VIN of (1) the dynamic latent transition kernel, as described in Section 3.1; (2) the network depth, as outlined in Section 3.2; (3) the adaptive highway loss, also covered in Section 3.2; and (4) the softmax function on the latent transition kernel, as mentioned in Section 3.2. Unless otherwise indicated, all these elements are present.

*Dynamic Latent Transition Kernel.* Figure 5(a) gives an illustration of DT-VIN's dynamic transition kernels and Figure 4(a) shows the SRs of our method with the proposed dynamic and the original invariant latent transition kernel. When using only the invariant transition kernel, a large drop in performance is observed. It is important to note here that the additional adaptive highway loss requires a high representational capacity of the latent MDP, meaning that removing the dynamic property of the kernel without removing the additional adaptive highway loss would be expected to adversely affect the performance of the original VIN. The dynamic transition kernel would be expected to be more beneficial in environments characterized by complex transitions—something common in advanced reinforcement learning domains. Indeed, as illustrated in Figure 5(b), the per-

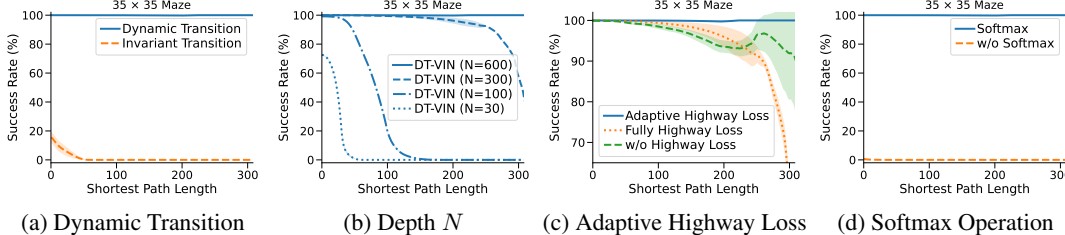

(a) Dynamic Transition     (b) Depth $N$     (c) Adaptive Highway Loss     (d) Softmax Operation

Figure 4: The results of ablation studies of our DT-VIN with 600 layers. (a) shows the performance of DT-VIN using a dynamic versus an invariant latent transition kernel. (b) shows the performance of DT-VIN over various depths of the planning module. (c) shows the performance of DT-VIN over different loss functions. (d) shows the performance of DT-VIN with and without the softmax operation on the latent transition kernel.

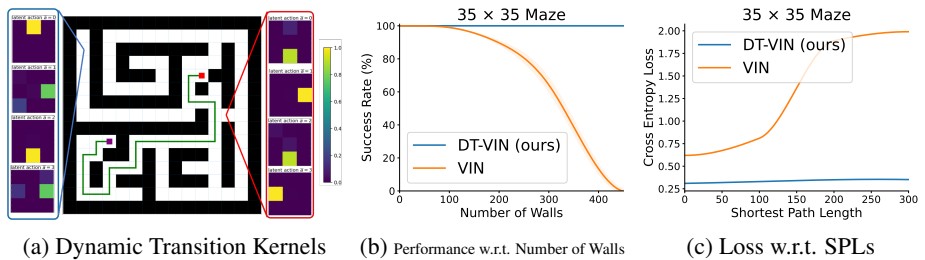

(a) Dynamic Transition Kernels     (b) Performance w.r.t. Number of Walls     (c) Loss w.r.t. SPLs

Figure 5: (a): Dynamic transition kernels of our DT-VIN, illustrated in a maze navigation example. The kernels on the left and right sides correspond to two distinct positions, respectively. Note that the size of latent action space is 4, resulting in 4 kernel matrices for each position. (b): Performance of our DT-VIN and VIN relative to the number of walls in the mazes. (c): Cross-entropy loss comparisons for the methods across tasks with varying SPLs.

formance gap between our DT-VIN and VIN grows when the number of walls increases. Likewise, the dynamic transition kernel is expected to be useful in long-term planning because it increases the representational capacity of the latent MDP. As demonstrated in Figure 5(c), this is the case here, with our DT-VIN exhibiting reduced compounding model errors over extended planning horizons when compared to the original VIN.

*Depth of Planning Module.* Figure 4(b) shows the SRs of our DT-VIN with various depths. Here, increasing the depth dramatically improves the long-term planning ability. For example, for tasks with an SPL of 200, the variant with depth $N = 300$ performs much better than the variant with depth $N = 100$. Moreover, for tasks with an SPL of 300, the deeper variant with depth $N = 600$ performs much better. Other methods like VIN and GPPN do not show a clear performance improvement when the depth increases. Figure 9 in Appendix C shows the performance of other methods over all depths.

*Adaptive Highway Loss.* We evaluate two variants of our DT-VIN, the first without the highway loss, and the second with a "fully highway loss," where the latter enforces a highway loss for each hidden layer without adaptive adjustment based on the actual planning steps. As shown in Figure 4(c), the variant without the highway loss suffers a decrease in performance, and the one with the fully highway loss performs even worse. These results imply that enforcing additional loss on hidden layers without any adjustment could harm performance. See Appendix C.4 for additional ablations on the highway loss components, the choice of the hyperparameter l, and the impact of the softmax operation on gradient stability.

*Softmax Latent Transition Kernel.* As shown in Figure 4(d), the variant without the softmax operation on the latent transition kernel fails on all the tasks. This failure is due to exploding gradients, wherein the gradient becomes extremely large, eventually resulting in the model's parameters overflowing and becoming a NaN (Not a Number) value.

## 4.2 3D VIZDOOM NAVIGATION

Following the methodology of the GPPN paper, we test our method on 3D ViZDoom (Wydmuch et al., 2019) environments. Here, instead of directly using the top-down 2D maze as in the previous experiments, we use the observation consists of RGB images capturing the first-person perspective of the environment, as illustrated in Figure 6(a). Then, a CNN is trained to predict the maze map from the first-person observation. The map is then given as input to the planning model, using the same architecture and hyperparameters as the 2D maze environments (see Appendix B.2 for more implementation details). For each algorithm, we select the best result across the various network depths $N = 30, 100, 300, 600$. We find that the optimal depth for DT-VIN is 600, for GPPN is 300, for VIN is 300, and for Highway VIN is 300. We evaluated the algorithm on 3D ViZDoom mazes with grid $35 \times 35$, where each cell in the grid corresponds to a $64 \times 64$ map unit area, the standard spatial measurement in the game engine. Figure 6(b) shows the SRs. Predictably, the performance of all the baselines decreases compared to the 2D maze environments due to the additional noise introduced by the predictions. Here, DT-VIN outperforms all the methods compared to the task over all the various SPLs.

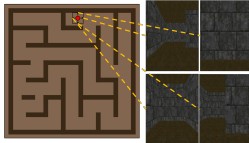

(a) 3D ViZDoom

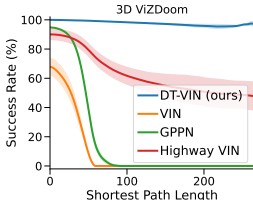

(b) Performance

Figure 6: (a) an example of a ViZDoom 3D maze and the first-person view of the environment with each of the corresponding four orientations. (b) the success rates of the algorithms over various SPLs.

## 4.3 ROVER NAVIGATION

We further evaluate the algorithms on the rover navigation task, where the algorithm must conduct planning based on an orthomosaic image (e.g., Figure 7(a)) created from aerial photographs. These images are usually more availible than elevation data (e.g., Figure 7(b)), which involve more complex processing using stereo image pairs (Goodchild, 2009). Therefore, we directly evaluate the path planning abilities of DT-VIN on orthomosaic images. We evaluate the Apollo 17 landing tasks, featuring images with a resolution of 0.5 meters per pixel, generated from images taken by the Lunar Reconnaissance Orbiter Camera's Narrow Angle Camera (Berlin, 2018). We crop the orthomosaic image into patches of various sizes, each $18 \times 18$ patch defining a cell. The expert paths are generated using external elevation data. The cell is considered a wall if the associated area exhibits an elevation angle exceeding 10 degrees. Note that elevation data are utilized solely for creating expert paths to train the algorithms and assessing algorithm performance, not as input to the neural networks. Please refer to Appendix B.3 for details on the task setting and the models' architecture. This task is challenging as orthomosaic image data does not typically include elevation information. Our DT-VIN outperforms all compared methods across various image sizes. Notably, with larger image sizes (particularly $630 \times 630$) our DT-VIN outperforms VIN by more than $5\%$. We also compare with an unfair baseline: CNN+$A^*$, which first trains a CNN to classify whether an $18 \times 18$ image patch is an obstacle with the elevation data and then use $A^*$ to conduct planning based on the prediction. While this unsuprisingly is able to outperform VIN, it still is itself outperformed by DT-VIN—despite DT-VIN being given access to only to expert trajectories and not to the elevation data (representing an overall weaker assumption).

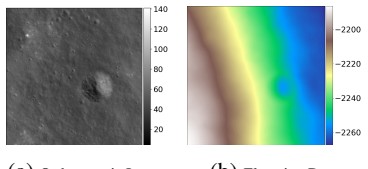

(a) Orthomosaic Image  (b) Elevation Data

| Training Method | Model | 270×270 | 450×450 | 630×630 |
|---|---|---|---|---|
| *without* elevation data | VIN | $85.32_{\pm 0.14}$ | $82.43_{\pm 0.74}$ | $71.71_{\pm 3.48}$ |
| | GPPN | $85.79_{\pm 0.31}$ | $81.72_{\pm 0.22}$ | $76.31_{\pm 0.75}$ |
| | Highway VIN | $85.81_{\pm 0.6}$ | $81.88_{\pm 0.86}$ | $73.21_{\pm 0.81}$ |
| | DT-VIN (ours) | $\mathbf{86.54_{\pm 0.5}}$ | $\mathbf{82.78_{\pm 0.6}}$ | $\mathbf{77.4_{\pm 0.98}}$ |
| *with* elevation data | CNN+$A^*$ | $84.42_{\pm 0.67}$ | $82.11_{\pm 0.74}$ | $76.19_{\pm 1.23}$ |

(c) The Sucess Rates on Tasks with Various Image Sizes

Figure 7: (a) and (b) shows a patch of orthomosaic image and elevation data from Apollo 17 landing tasks. (c) lists the success rates of the algorithms on rover navigation tasks with various image sizes.

## 5 RELATED WORK

**Variants of Value Iteration Networks (VINs).** Several variants of VIN (Tamar et al., 2016) have been proposed in recent years. Gated Path Planning Networks employ gating recurrent mechanisms to reduce the training instability and hyperparameter sensitivity seen in VINs (Lee et al., 2018). To mitigate overestimation bias (which is detrimental to learning here), dVINs were proposed and use a weighted double estimator as an alternative to the maximum operator (Jin et al., 2021). For addressing challenges in irregular spatial graphs, Generalized VINs adopt a graph convolution operator, extending the traditional convolution operator used in VINs (Niu et al., 2018). To improve scalability, AVINs introduce an abstraction module that extracts higher-level information from the environment and the goal (Schleich et al., 2019). For transfer learning, Transfer VINs address the generalization of VINs to target domains where the action space or the environment's features differ from those of the training environments (Shen et al., 2020). More recently, VIRN was proposed and employs larger convolutional kernels to plan using fewer iterations as well as self-attention to propagate information from each layer to the final output of the network (Cai et al., 2022). Similarly, GS-VIN also uses larger convolutional kernels but to stabilize training and also incorporates a gated summarization module that reduces the accumulated errors during value iteration (Cai et al., 2023). Most related to DT-VIN is other recent work that focused on developing very deep VINs for long-term planning. Specifically, Highway VIN (Wang et al., 2024a) incorporates the theory of Highway Reinforcement Learning (Wang et al., 2024b) to create deep planning networks with up to 300 layers for long-term planning tasks. Highway VIN modifies the planning module of VIN by introducing an exploration module that injects stochasticity in the forward pass and uses gating mechanisms to allow selective information flow through the network layers. Our method, however, achieves even deeper planning by incorporating a dynamic transition matrix in the latent MDP and adaptively weighting each layer's connection to the final output.

**Neural Networks with Deep Architectures.** There is a long history of developing very deep neural networks (NNs). For sequential data, this prominently includes the LSTM architecture and its gated residual connections, which help alleviate the "vanishing gradient problem" (Hochreiter & Schmidhuber, 1997; Hochreiter, 1991). For feedforward NNs, a similar gated residual connection architecture was used in Highway Networks (Srivastava et al., 2015a) and later in the ResNet architecture (He et al., 2016), where the gates were kept open. Such residual connections are still ubiquitous in modern language architectures, such as the Generative Pre-trained Transformer (GPT) (Achiam et al., 2023). Our method dynamically employs skip connections from select hidden layers to the final loss, utilizing a state and observation map-dependent transition kernel. This approach is more closely aligned with the computation of the true VI algorithm. Similar kernels, dependent on an input image (Chen et al., 2020) or the coordinates of an image (Liu et al., 2018), have been previously used in Computer Vision.

## 6 CONCLUSIONS

Planning is a long-standing challenge in the field of artificial intelligence and its subfield: reinforcement learning. Previous work proposed VIN as an end-to-end differentiable neural network architecture for this task. While VINs have been successful at short-term small-scale planning, they start to fail quite rapidly as the horizon and the scale of the planning grows. We observed that this decay in performance is principally due to limitations in the (1) representational capacity of their network and (2) its depth. To alleviate these problems, we propose several modifications to the architecture, including a dynamic transition kernel to increase the representation capacity and an adaptive highway loss function to ease the training of very deep models. Altogether, these modifications have allowed us to train networks with 5000 layers. In line with previous work, we evaluate the efficacy of our proposed Dynamic Transition VINs (DT-VINs) on 2D maze, 3D ViZDoom, and rover navigation environments. We find that DT-VINs scale to longer-term and larger-scale planning problems than previous attempts. To the best of our knowledge, DT-VINs is, at the time of publication, the current state-of-the-art planning solution for these specific environments. We note that the upper bound for this approach (i.e., the scale of the network and, consequentially, the scale of the planning ability) remains unknown. As our experiments were limited mostly by computational cost and did not observe instability, we expect that with the growth of available computational power, our method will scale to even longer-term and larger-scale planning.

## REPRODUCIBILITY STATEMENT

The details of model architecture, along with specific training protocols, can be found in Section 4 and Appendix B. We will open-source the code for reproducing the results of this paper after publication.

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

## A LIMITATIONS AND FUTURE WORKS

The principal limitation of our work compared to VIN and Highway VIN is the increased computational cost (see Appendix C.5). This is a consequence of the scale of the network. The past decades have seen AI dominated by the trend of scaling up systems (Sutton), so this is not likely a long-term issue. Other limitations include the requirement to know the length of the shortest path $l$ in the highway loss in imitation learning. In a general RL problem, such a quantity could be estimated online. Future work will explore the impact of a more sophisticated transition mapping module (this work uses a single CNN layer for this purpose) in more challenging real-world applications, such as real-time robotics navigation in dynamic and unpredictable environments.

## B EXPERIMENTAL DETAILS

The below subsections detail specific information about the experiments that have been deemed too minor to appear in the main text.

### B.1 2D MAZE NAVIGATION

Figure 8 shows some visualizations of some of the different 2D maze navigation tasks we experiment with. Our experimental setup follows the guidelines established in the GPPN paper (Lee et al., 2018). For these tasks, the datasets for training, validation, and testing comprise 25000, 5000, and 5000 mazes, respectively. Each maze features a goal position, with all reachable positions selected as potential starting points. Note that this setting, as done by GPPN, produces a distribution of mazes with non-uniform SPLs, which is skewed towards shorter SPLs. Table 3 shows the hyperparameters used by our method. Note that, while DT-VIN consistently uses 3 for the size of the latent transition kernel $F$ and 4 for the size of the latent action space $|\overline{\mathcal{A}}|$, other methods instead used their best-performing sizes from between 3 and 5, and between 4 and 150, respectively.

Moreover, to reduce the computational complexity of highway loss, we apply adaptive highway loss only to layers $n$ satisfying the condition $n \bmod J = 0$, where $J$ is a hyperparameter set to 10 in our experiments. Here, the main idea is to build the highway connections at interval $J$, for example, every 10 neural network layers. Using this, the number of the loss terms will reduce to only $1/J$ of the original one. Table 2 shows the magnitude of the computational speedup as a concequence of this implementation detail.

| | Wall-Clock Time (hours) | [1,100] | [100,200] | [200,300] |
|---|---|---|---|---|
| $J = 1$ | 37 | $\mathbf{100.00_{\pm 0.00}}$ | $\mathbf{99.99_{\pm 0.01}}$ | $\mathbf{99.78_{\pm 0.21}}$ |
| $J = 10$ | 12.1 | $\mathbf{100.00_{\pm 0.00}}$ | $\mathbf{99.99_{\pm 0.01}}$ | $99.77_{\pm 0.23}$ |
| $J = 50$ | 7.1 | $100.00_{\pm 0.00}$ | $99.98_{\pm 0.02}$ | $99.69_{\pm 0.27}$ |

Table 2: Training Time and Success Rate (%) across Different Ranges of SPLs for DT-VIN with Different $J$ Values.

### B.2 3D VIZDOOM

To be in line with previous work, we use a state representation preprocessing stage for the 3D ViZDoom environment similar to that used in the GPPN paper and others (Lee et al., 2018; Lample & Chaplot, 2017). In 3D ViZDoom, a maze is designed on a grid of $M \times M$ cells. Each cell in this grid corresponds to an area of $64 \times 64$ map units within the 3D ViZDoom environment. The map unit is the basic measure of space used in the ViZDoom game engine to define distances and sizes. Specifically, for each cell in the $M \times M$ 3D maze, the RGB first-person views for each of the four cardinal directions are given as state to a preprocessing network (see Figure 6(a)). This network then encodes this state and produces an $M \times M$ binary maze matrix. The hyperparameters and exact specification of the network are given in Table 5.

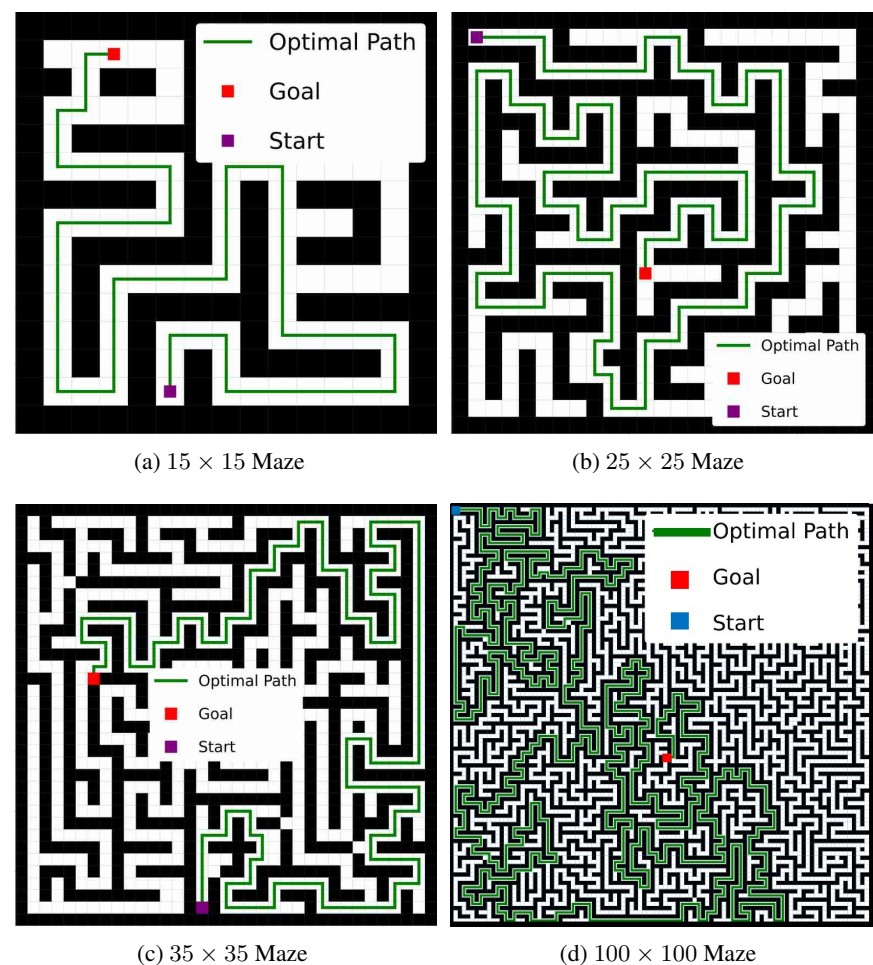

(a) $15 \times 15$ Maze

(b) $25 \times 25$ Maze

(c) $35 \times 35$ Maze

(d) $100 \times 100$ Maze

Figure 8: Some examples of the 2D maze navigation tasks.

Table 3: 2D Maze Navigation Hyperparameters

| Hyperparameter | Value |
|---|---|
| Transition Mapping Module | Conv with $3 \times 3$ kernel |
| Reward Mapping Module | Conv with $1 \times 1$ kernel |
| Latent Transition Kernel Size ($F$) | 3 |
| Latent Action Space Size ($|\overline{\mathcal{A}}|$) | 4 |
| Optimizer | RMSprop |
| Learning Rate | 1e-3 |
| Batch Size | 32 |
| Depth of Planning Module | $15 \times 15$ maze: 200
$25 \times 25$ maze: 200
$35 \times 35$ maze: 600
$100 \times 100$ maze: 5000 |

## B.3 ROVER NAVIGATION

Table 4 shows the hyperparameters of DT-VIN for the rover navigation tasks. For the transition and reward mapping modules, we employ 10-layer CNNs, with the first 8 layers shared between them.

Table 4: Rover Navigation Hyperparameters

| Hyperparameter | Value |
|---|---|
| Transition Mapping Module | A 10-layer CNN |
| Reward Mapping Module | A 10-layer CNN (sharing the first 8 layers with Transition Mapping Module) |
| Latent Transition Kernel Size ($F$) | 3 |
| Latent Action Space Size ($|\overline{\mathcal{A}}|$) | 4 |
| Optimizer | RMSprop |
| Learning Rate | 1e-3 |
| Batch Size | 32 |
| Depth of Planning Module | $270 \times 270 : 50$ $450 \times 450 : 100$ $630 \times 630: 200$ |

Table 5: 3D ViZDoom Preprocessing Network

| Hyperparameter | Value |
|---|---|
| Batch Size ($B$) | 32 |
| Image Directions ($D$) | 4 |
| Image Channels ($C$) | 3 |
| Image Width ($W$) | 24 |
| Image Height ($H$) | 32 |
| Input Size | $(B, M, M, D, W, H, C)$ |
| Layer 1 (Convolution) | $(3, 32, 8, 4, 1)$ |
| Layer 2 (Convolution) | $(32, 64, 4, 2, 1)$ |
| Layer 3 (Linear) | $(384, 256)$ |
| Layer 4 (Convolution) | $(1024, 64, 3, 1, 1)$ |
| Layer 5 (Convolution) | $(64, 1, 3, 1, 1)$ |
| Output Size | $(B, M, M)$ |
| Optimizer | Adam |
| Learning Rate | 1e-3 |
| Betas | $(0.9, 0.999)$ |

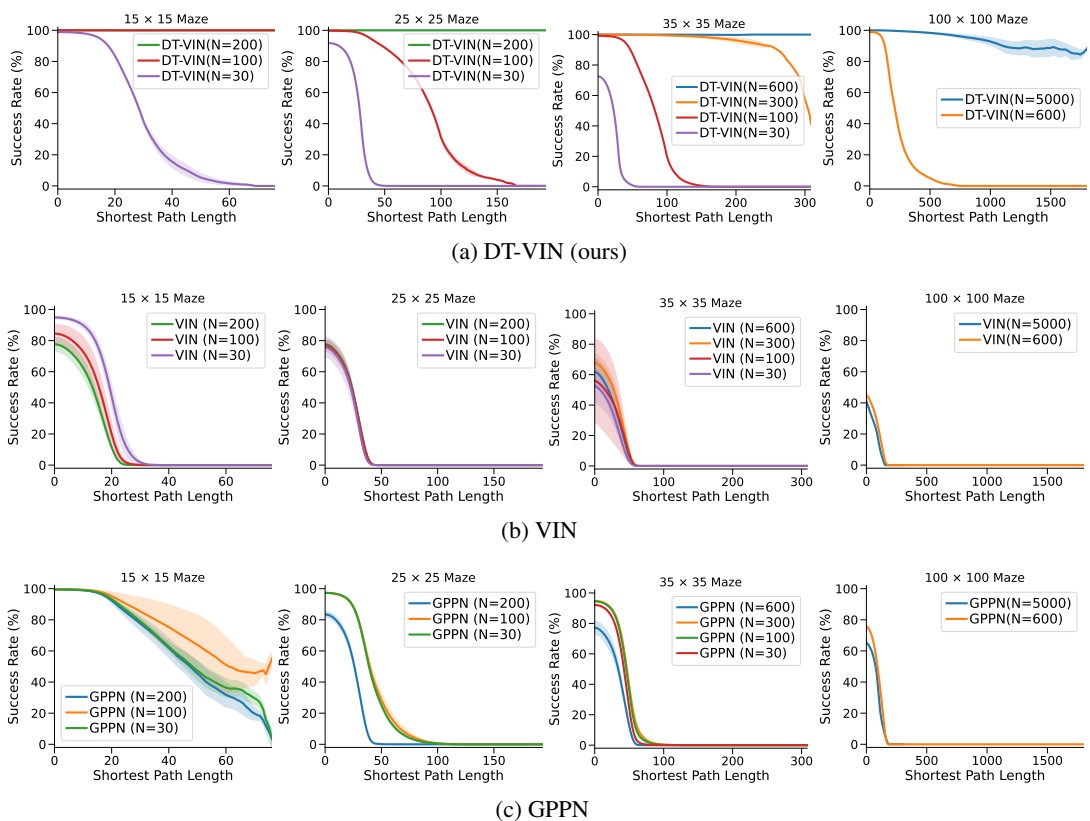

Figure 9: The success rate of each method as a function of shortest path length and network depth. The green and red curves overlap in the first plot of (a).

## C    ADDITIONAL EXPERIMENTAL RESULTS

Due to space constraints, the below results could not appear in the main text.

### C.1    PERFORMANCE OF MODELS ACROSS DIFFERENT DEPTHS

Figure 9 shows the success rate of all the algorithms on the $15 \times 15$, $25 \times 25$, $35 \times 35$, $100 \times 100$ mazes as a function of the shortest path length and the depth of the network. Similarly, Figure 10 shows the corresponding optimality rates.

### C.2    DIFFERENT TRANSITION KERNELS

Following the GPPN paper (Lee et al., 2018), We have run an additional ablation using different transition kernels: the *Differential Drive* transition kernel, where the agent can move forward along its orientation or rotate 90 degrees left or right, and the *MOORE* transition kernel, where the agent can relocate to any of the eight adjacent cells that comprise its Moore neighborhood. As shown in Table 6 and 7, DT-VIN consistently outperforms all the compared methods regardless of the kernel used.

### C.3    EXPERIMENTS ON CONTINUOUS CONTROL

To further demonstrate the generalizability of DT-VIN to different domains, we compare its performance to VIN, GPPN, and Highway VIN on Point Maze (He et al., 2016), a continuous control

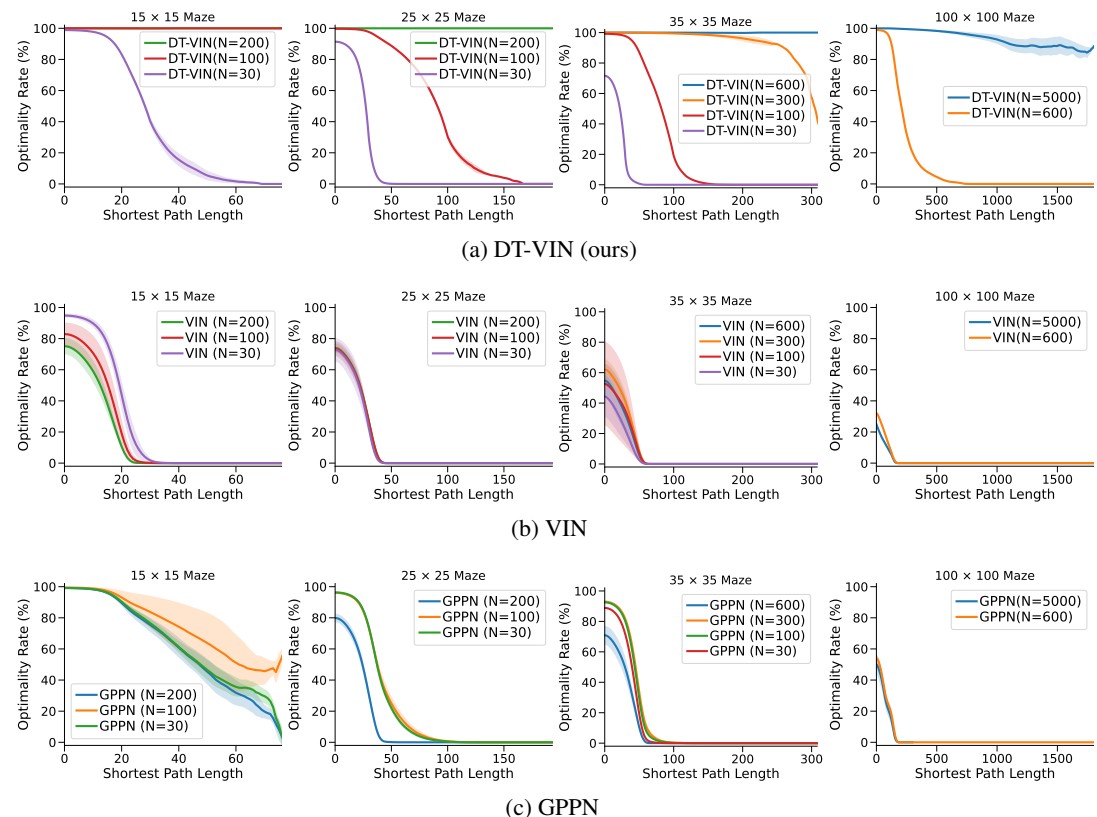

Figure 10: The optimality rate of each method as a function of shortest path length and network depth. The green and red curves overlap in the top-left plot.

Table 6: The success rate (%) for each method in $35 \times 35$ 2D maze navigation with *Differential Drive* transition kernel, where the agent can move forward along its orientation or rotate $90°$ left or right.

| Shortest Path Length | [1,150] | [150,300] | [300,500] |
|---|---|---|---|
| VIN | $68.44_{\pm 3.12}$ | $0.03_{\pm 0.01}$ | $0.00_{\pm 0.00}$ |
| GPPN | $83.1_{\pm 1.23}$ | $0.31_{\pm 0.01}$ | $0.0_{\pm 0.0}$ |
| Highway VIN | $87.1_{\pm 3.73}$ | $57.1_{\pm 3.98}$ | $49.1_{\pm 8.73}$ |
| DT-VIN (ours) | $\mathbf{100.00_{\pm 0.00}}$ | $\mathbf{100.00_{\pm 0.00}}$ | $\mathbf{99.99_{\pm 0.01}}$ |

domain. Here, as shown in Figure 11, the agent needs to apply force to a ball to navigate a maze and reach the goal within 800 steps. Table 8 shows the results of this experiment. Again, DT-VIN is able to solve the mazes at a much higher rate than all the baseline methods and typically ends episodes with the ball much closer to the goal.

### C.4 ABLATION ON SOFTMAX OPERATION AND HIGHWAY LOSS

**Ablation on Highway Loss**   We conduct an ablation study for the adaptive highway loss by evaluating the following variants in shorter planning tasks:

1. Implement skip connections for intermediate layers of the planning module of VIN, like what has been done in Residual Nets (He et al., 2015) and Highway Nets (Srivastava et al., 2015b). As shown in Table 9, this variant performs poorly, achieving only 61.35% success rate in comparison to DT-VIN's 99.98% on $100 \times 100$ Maze. These results are consistent with those in existing work (Wang et al., 2024a).

Table 7: The success rate for each method in $35 \times 35$ 2D maze navigation with *Moore* transition kernel, where the agent can relocate to any of the eight adjacent cells that comprise its Moore neighborhood.

| Shortest Path Length | [1,100] | [100,200] | [200,250] |
|---|---|---|---|
| VIN | $66.44_{\pm 3.21}$ | $0.00_{\pm 0.00}$ | $0.00_{\pm 0.00}$ |
| GPPN | $89.94_{\pm 1.31}$ | $0.04_{\pm 0.01}$ | $0.00_{\pm 0.00}$ |
| Highway VIN | $83.14_{\pm 2.21}$ | $37.1_{\pm 1.98}$ | $25.1_{\pm 3.28}$ |
| DT-VIN (ours) | $\mathbf{100.0_{\pm 0.00}}$ | $\mathbf{98.9_{\pm 0.72}}$ | $\mathbf{96.7_{\pm 1.23}}$ |

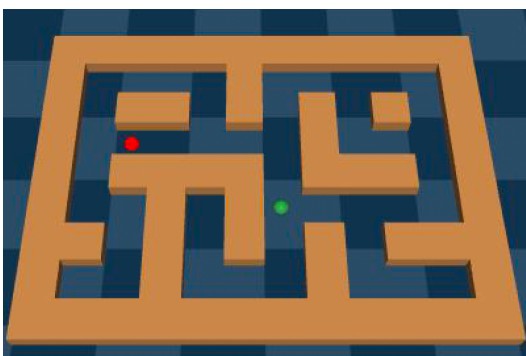

Figure 11: The Point Maze environment (He et al., 2016).

Table 8: Comparison of Final Distance to the Goal and Success Rate for Different Models in the Continuous Control Setting.

| | Final Distance to the Goal (Euclidean Distance) | Success Rate (%) |
|---|---|---|
| VIN | $5.12 \pm 3.19$ | $62.00 \pm 2.18$ |
| GPPN | $4.12 \pm 2.18$ | $68.12 \pm 4.17$ |
| Highway VIN | $4.98 \pm 3.28$ | $67.31 \pm 3.28$ |
| DT-VIN (ours) | $\mathbf{2.28 \pm 1.20}$ | $\mathbf{82.00 \pm 3.89}$ |

2. $\mathbf{1}_{\{n=l\}}$, only building highway loss for a specific layer $n$ which satisfies $n = l$. As shown in Table 9, this variant performs worse than the adaptive highway loss, showing that the component $n > l$ plays an important role in the performance.

3. Building highway loss for all intermediate layers $n$, without the term $\mathbf{1}_{\{n \geq l\}}$. This variant is already verified to be less effective in Figure 4(c).

Table 9: Success rates for the variants of adaptive highway loss on 2D Maze.

| Method | $35 \times 35$, **SPL [1,100]** | $100 \times 100$, **SPL [1,600]** |
|---|---|---|
| Skip Connections for intermediate layers | $90.35_{\pm 2.53}$ | $61.35_{\pm 3.43}$ |
| $\mathbf{1}_{\{n \geq l\}}$ (Adaptive Highway Loss) | $\mathbf{100.00_{\pm 0.00}}$ | $\mathbf{99.98_{\pm 0.00}}$ |
| $\mathbf{1}_{\{n=l\}}$ (without $\mathbf{1}_{\{n \geq l\}}$) | $98.35_{\pm 2.23}$ | $92.81_{\pm 3.78}$ |
| Without $\mathbf{1}_{\{n \geq l\}}$ (Fully Highway Loss) | $98.11_{\pm 1.23}$ | $91.11_{\pm 2.00}$ |

**Gradient and Loss Analysis** The softmax operation ensures that the values of the dynamic transition kernels remain within $[0, 1]$, helping to prevent the gradient exploding problem. In our experiments, we found that the gradient of DT-VIN lacking softmax operation explodes at the first forward-backward pass of training, resulting in the loss escalating to NaN (Not a Number) during the training process. Figure 12 shows the gradient and the loss of DT-VIN with and without Softmax Operation.

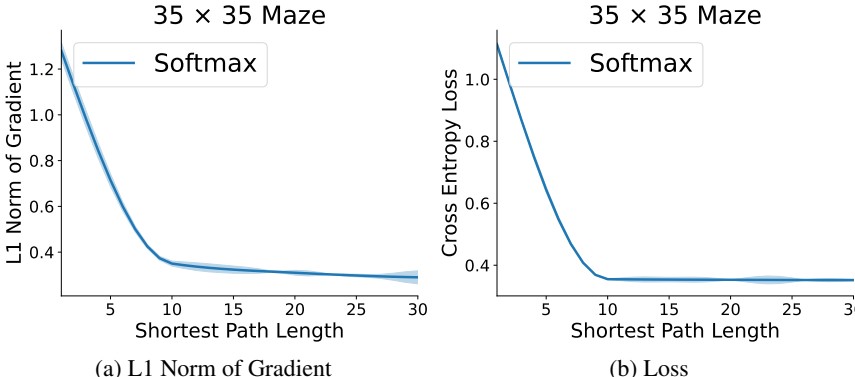

(a) L1 Norm of Gradient          (b) Loss

Figure 12: The L1 norm of gradient averaged over the first 10 layers and the loss during the training process for DT-VIN with Softmax Operation, evaluated on $35 \times 35$ 2D maze with depth $N = 600$. The result of DT-VIN without softmax operation is missing, as the gradient explodes at the first forward-backward pass of training, resulting in the loss escalating to NaN (Not a Number) during the training process.

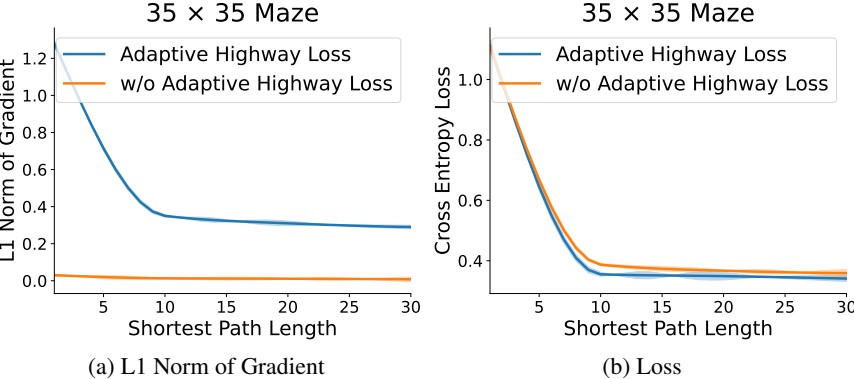

(a) L1 Norm of Gradient          (b) Loss

Figure 13: The L1 norm of gradient averaged over the first 10 layers and the loss during the training process for DT-VIN with and without Adaptive Highway Loss, evaluated on $35 \times 35$ 2D maze with depth $N = 600$.

The adaptive highway loss improves gradient flow toward shallower layers. As shown in Figure 13, without adaptive highway loss, the L1 norm of the gradient for DT-VIN is closer to zero in the first 10 layers of the network. The adaptive highway loss can reduce this vanishing gradient problem, resulting in lower loss.

**Ablation on the Choice of** $l$     The knowledge of the length $l$ of the expert path naturally exists in the imitation learning case. However, for the case where such information is unknown, one can use either the length of non-expert data or some heuristic methods to estimate $l$ when the actual $l$ is completely unknown, e.g., using the distance between the start and the goal position.

To measure the effect of overestimation/underestimation, we experiment with various estimated values of the length of the shortest path $\hat{l}$, which are $0, l/2, l, 2l, N$ (where $l$ is the actual length of the shortest path, $N$ is the depth of the planning module). Second, to evaluate the case when the etsimation of $l$ has variance, we use $l \cdot \max(\epsilon, 0)$ as the estimation, with $\epsilon$ sampled from a Gaussian distribution $\mathcal{N}(1, 1)$. Third, we also assess two additional variants for estimating $l$: (a) One variant that utilizes the length of non-expert trajectories for $l$; (b) Another variant that estimates the shortest path length heuristically using the L1 distance between the start $(x_s, y_s)$ and the goal $(x_g, y_g)$, i.e., $D = |x_s - x_g| + |y_s - y_g|$.

As indicated in Table 10, both overestimation and underestimation lead to a performance degradation of no more than $7\%$. Additionally, we find that leveraging non-expert data or the heuristic L1

distance only yields a nearly 3% degradation in performance, and performs better than the case when the optimal length is extremely overestimated/underestimated. These resutls imply that employing the information from non-expert data or heuristic estimation could be taken as an alternative when the optimal length is not available.

Table 10: Ablation study for using various estimated lengths of optimal paths for adaptive highway loss, under $35 \times 35$ ViZDoom navigation. The best results are highlighted.

| Shortest Path Length | [1,100] | [100, 200] | [200,300] |
|---|---|---|---|
| $\widehat{l} = 0$ (connected to all hidden layers) | $99.49_{\pm 0.35}$ | $94.51_{\pm 0.77}$ | $89.1_{\pm 3.56}$ |
| $\widehat{l} = l/2$ | $99.62_{\pm 0.91}$ | $96.21_{\pm 0.44}$ | $91.24_{\pm 1.68}$ |
| $\widehat{l} = l$ | $\mathbf{99.67_{\pm 0.22}}$ | $\mathbf{97.92_{\pm 0.11}}$ | $\mathbf{96.41_{\pm 0.37}}$ |
| $\widehat{l} = 2 * l$ | $99.61_{\pm 0.18}$ | $96.29_{\pm 0.48}$ | $93.12_{\pm 0.73}$ |
| $\widehat{l} = N$ (connected to only last layer) | $99.52_{\pm 0.29}$ | $95.52_{\pm 0.86}$ | $91.12_{\pm 1.64}$ |
| $\widehat{l} = l \cdot \max(\epsilon, 0), \epsilon \sim \mathcal{N}(1, 1)$ | $99.62_{\pm 0.50}$ | $96.19_{\pm 0.15}$ | $93.21_{\pm 0.92}$ |
| $\widehat{l} = len(\text{non-expert path})$ | $99.62_{\pm 0.12}$ | $97.01_{\pm 0.69}$ | $93.31_{\pm 0.31}$ |
| $\widehat{l} = D$ (L1 distance) | $99.64_{\pm 0.49}$ | $96.92_{\pm 0.05}$ | $93.52_{\pm 0.87}$ |

## C.5 SCALING EXPERIMENTS

**Compute** As we have discussed in Section 3.1, our approaches only require $|\overline{\mathcal{A}}| \times F^4$ parameters, where we set $|\overline{\mathcal{A}}| = 4$ and $F = 3$ in our experiments. Table 12 shows the memory consumption and training time on NVIDIA A100 GPUs for DT-VIN and the compared methods when using 5000 layers and training for 90 epochs on $100 \times 100$ maze. As shown in the table below, our DT-VIN consumes significantly less GPU memory compared to GPPN, while requiring a similar amount of GPU hours. These results are generally consistent with those observed in the $35 \times 35$ 2D maze in Table 11.

Table 11: The computational complexity during traning of each method, employing 600 layers and trained over 30 epochs, evaluated in a $35 \times 35$ 2D maze navigation.

| Method | GPU Memory (GB) | Wall-Clock Time (h) | GPU Hours (h) |
|---|---|---|---|
| VIN | 4.2 | 8.4 | 8.4 |
| GPPN | 182 | 4.2 | 12.6 |
| Highway VIN | 41.3 | 14.3 | 14.3 |
| DT-VIN | 53.3 | 12.1 | 12.1 |

Table 12: Computational complexity during of training of each method using 5000 layers and training for 90 epochs, evaluated on a $100 \times 100$ 2D maze navigation.

| Method | GPU Memory (GB) | Wall-Clock Time (h) | GPU Hours (h) |
|---|---|---|---|
| VIN | 35 | 36 | 36 |
| GPPN | 710 | 31 | 310 |
| Highway VIN | 111 | 112 | 224 |
| DT-VIN (ours) | 182 | 98 | 294 |

**Model size** In our experiments, the depth of the network required to solve the problem is close to linear with the number of planning steps required by the problem. For maze size $M = 15, 25, 35$, we test DT-VIN models at increasing depths in increments of 100 until the optimal performance is achieved. For instance, for mazes of size $25 \times 25$, we assess depths of $100, 200, 300, 400$. For maze size $M = 100$, we assess depths of $4000, 5000, 6000$. As Table 13 illustrates, the depth of the smallest network that can solve the task increases slightly more than linearly with the required planning steps. Therefore, it might be feasible to continue increasing the network depth as the problems become more complex.

Table 13: Minimal depths of DT-VIN model across various maze sizes.

| Maze Size | Longest Length of Optimal Path | Minimal Depth of DT-VIN |
|---|---|---|
| 15 | 80 | 100 |
| 25 | 200 | 300 |
| 35 | 300 | 500 |
| 100 | 1800 | 5000 |

**Data** The scale of the dataset needs to scale up with the complexity of the problem rather than the model depth. Under the same scale of the problem, we didn't find that increasing model depth requires additional data. As shown in Table 14, without expanding the dataset, increasing the model depth does not reduce the performance.

Moreover, even in situations where data is rare, DT-VIN still outperforms compared methods. As shown in Table 15, with only $50\%$ of the original dataset, DT-VIN greatly outperforms existing methods. We also highlight the changes compared to the performance with a full-sized dataset in Table 16, where DT-VIN results in less than a $0.2\%$ degradation for tasks within the range $[1, 100]$, while the best-performing comparison method, GPPN, incurs a degradation of nearly $12\%$.

Table 14: The success rate of DT-VIN across various model depths $N$, maintaining the same size as the original dataset.

| Shortest Path Length | [1,100] | [100,200] | [200,300] |
|---|---|---|---|
| $N = 300$ | $99.99_{\pm0.01}$ | $99.81_{\pm0.13}$ | $92.11_{\pm1.31}$ |
| $N = 600$ | $100.00_{\pm0.00}$ | $99.99_{\pm0.01}$ | $99.77_{\pm0.23}$ |
| $N = 1200$ | $100.00_{\pm0.00}$ | $99.99_{\pm0.01}$ | $99.81_{\pm0.11}$ |

Table 15: The success rate for each method, using a dataset reduced to **50%** of the original size.

| Shortest Path Length | [1,100] | [100,200] | [200,300] |
|---|---|---|---|
| VIN | $32.41_{\pm4.25}$ | $0.00_{\pm0.00}$ | $0.00_{\pm0.00}$ |
| GPPN | $83.11_{\pm1.33}$ | $0.01_{\pm0.01}$ | $0.00_{\pm0.00}$ |
| Highway VIN | $45.41_{\pm4.13}$ | $37.41_{\pm3.25}$ | $21.41_{\pm6.98}$ |
| DT-VIN (ours) | $\mathbf{99.96_{\pm0.01}}$ | $\mathbf{99.8_{\pm0.12}}$ | $\mathbf{96.01_{\pm0.32}}$ |

Table 16: The *changes* in success rate for each method, using a dataset reduced to **50%** of the original size, compared to the full-sized dataset (more negative is worse).

| Shortest Path Length | [1,100] | [100,200] | [200,300] |
|---|---|---|---|
| VIN | $-36.00_{\pm3.12}$ | $0.00_{\pm0.00}$ | $0.00_{\pm0.00}$ |
| GPPN | $-12.60_{\pm1.29}$ | $-0.38_{\pm0.11}$ | $0.00_{\pm0.00}$ |
| Highway VIN | $-45.26_{\pm3.48}$ | $-28.09_{\pm2.98}$ | $-32.99_{\pm3.11}$ |
| DT-VIN (ours) | $\mathbf{-0.04_{\pm0.01}}$ | $\mathbf{-0.19_{\pm0.04}}$ | $\mathbf{-3.76_{\pm0.31}}$ |

