# OpenReview forum: "Scaling Value Iteration Networks to 5000 Layers for Extreme Long-Term Planning"
_ICLR.cc/2025/Conference — Submitted to ICLR 2025_

### Official Review · Reviewer_FzU7 · 2024-10-26

**Soundness:** 2
**Presentation:** 3
**Contribution:** 2
**Rating:** 5
**Confidence:** 3

**Summary:**

The paper extends the Value Iteration Networks (VIN) to handle long-term planning by making two modifications: 1. Changing the implementation of latent transition kernel in VIN from state-invariant to observation-dependent; 2. Increasing the depth of the planning module with the “Highway” connections which are defined based on the optimal planning path. The proposed algorithm, named DT-VIN, has been evaluated on 2D maze navigation tasks, a converted 3D VizDoom task and a rover navigation task with orthomosaic image as input. The empirical evaluation shows DT-VIN performs generally well, especially on large-size mazes and deep planning modules.

**Strengths:**

* [**Originality**] The paper considers a rather deep planning module, comprising of many value iteration modules and shows that such deep architecture is required to solve complex maze tasks.
* [**Quality**] The proposed method has been evaluated on various benchmark tasks and the performance is generally better.
* [**Clarity**] The paper is well written, and most parts are easily understandable. Figure 2 presents a nice overview of the main idea of the proposed method. Figures in ablation studies also clearly shows the ablation effects of each design consideration.
* [**Significance**] The paper shows that solving a long-term planning requires a deeper planning module architecture, which can be a good contribution to the community.

**Weaknesses:**

### Originality: VIN can be observation-dependent
One of the main contributions claimed by this paper is the incorporation of observation into the transition kernel. However, as mentioned by the paper itself, **“original VIN paper proposes a general framework where the latent transition kernel depends on the observation”**. In other words, the VIN has incorporated the observation into the learning of transition kernel already, but its open-sourced implementation does not reflect this. So this can hardly be regarded as a contribution of this paper. **The claim of incorporation of observation may sound like a good implementation suggestion when the considered the maze is of bigger sizes**.

### Significance: additional loss introduced by highway connections
The paper introduces the highway connections and thus incur extra loss to the final loss. Such extra loss relies on two important factors: the length of the optimal path (the provided expert path from the start to the goal) and the number of layers for the planning module. **It would be important to discuss the situations where the optimal path might not be known in advance**. For example, in the VizDoom experiment, it is unclear to me how such optimal path is obtained. Further, the number of layers has a big impact on the loss scales. Consider the extreme case of 5000 layers, if the optimal path is 1000, then the final loss will have 4000+1 terms, which can make the optimization hard as the effective backpropagation of gradients can be challenging. **The paper should discuss how this long-range backpropagation of gradients can be well handled in practise**.

**Questions:**

1. Since the training of highway connections requires the expert path from the start to the goal, how the training and testing mazes are constructed in the empirical evaluation? If the training and testing mazes are the same, the extra loss might leak the path info to the planning module.
2. In Figure 9, there is no plot of 100x100 maze. Why is such plot not provided? Given that the depth of 5000 is required only for such maze, it would be crucial to see how the choice of other depth values would affect the SR and OR.
3. The description to VizDoom experiment is vague and more details are needed. Can the authors clarify how the optimal path can be found and what it means for “3D ViZDoom mazes with size M = 35”?
4. The evaluation on ROVER NAVIGATION is also unclear. The paper says that “elevation data are utilized solely for creating expert paths and assessing algorithm performance, not as input for the neural networks”. But the expert paths are needed to define the training loss (i.e., the additional loss), right? Then the proposed method DT-VIN still (implicitly) requires the evaluation data as the input?

---

> ### Author Response · Authors · 2024-11-24
> **Response to Reviewer FzU7 (1/4)**
>
> We appreciate the valuable comments and advice from Reviewer FzU7. Below, we provide detailed responses to the reviewer's questions.
>
>
> ## Weaknesses
>
> ---
>  ### W1:
> >  One of the main contributions claimed by this paper is the incorporation of observation into the transition kernel. However, as mentioned by the paper itself, “original VIN paper proposes a general framework where the latent transition kernel depends on the observation”. In other words, the VIN has incorporated the observation into the learning of transition kernel already, but its open-sourced implementation does not reflect this. So this can hardly be regarded as a contribution of this paper. The claim of incorporation of observation may sound like a good implementation suggestion when the considered the maze is of bigger sizes.
>
> ### Answer:
>
>
> There is some depth to the question of whether, proposing a method with a general definition but examining only a subset of that definition (e.g., invariant transition), allows another to claim a contribution by recognizing the importance of a different subset (e.g., observation-dependent transition).
> We would argue that the answer to this is yes.
> Indeed, we base our contribution on discovering that this specific version, i.e., observation-dependent transitions, allows for addressing scalability challenges not tackled by the original VIN paper.
> Moreover, the original VIN paper did not specify whether one should employ fully connected or convolutional layers for the transition mapping. In contrast, we advocate for the use of a single convolutional layer, which proves to be effective while adding minimal computational overhead.
> Altogether, we endeavour to be fair to the original work and thus acknowledge it here, but it does seem that we found something very important that was simply missed by the original VIN work: the specific variations that allow it to solve long-term planning.
> We believe this increases the practical applicability and robustness of the model, making it a substantial contribution beyond the original framework's scope. This adaptation opens new avenues for applying VIN in more diverse and challenging scenarios, which we have demonstrated with empirical results in our paper.

---

> > ### Comment · Reviewer_FzU7 · 2024-11-27
> > **Thank you for the response**
> >
> > Thanks for the clarification. I think the question is different here: first the authors and the reviewer agreed that "the VIN has incorporated the observation into the learning of transition kernel already", which is not novel. Second, re the architecture change from fully connected to convolutional layers, isn't it also coming from the VIN paper? as "CNN-based VIN uses an invariant latent transition kernel"?

---

> ### Author Response · Authors · 2024-11-24
> **Response to Reviewer FzU7 (2/4)**
>
> ---
>  ### W2:
> >  ...highway loss relies on two important factors: the length of the optimal path (the provided expert path from the start to the goal) and the number of layers for the planning module. It would be important to discuss the situations where the optimal path might not be known in advance. For example, in the VizDoom experiment, it is unclear to me how such optimal path is obtained.
>
> ### Answer:
>
>
>
> Thanks for this insightful comment.
> Note that the knowledge of the length $l$ of the expert path naturally exists in the imitation learning case.
> However, for the case where such information is unknown, one can use either the length of non-expert data or some heuristic methods to estimate $l$ when the actual $l$ is completely unknown, e.g., using the distance between the start and the goal position.
> To further address your comment, we have now run a number of ablations on this.
>
>
>
>
> First, to measure the effect of overestimation/underestimation, we conducted the experiment with various estimated values of the length of the shortest path $\widehat{l}$, which are $0, l/2, l, 2l, N$ (where $l$ is the actual length of the shortest path, $N$ is the depth of the planning module).
> Second, to evaluate the case when the estimation of $l$ has variance, we use $l \cdot \max(\epsilon, 0)$ as the estimation, with $\epsilon$ sampled from a Gaussian distribution $\mathcal{N}(1, 1)$.
> Third, we also assessed two additional variants for estimating $l$:
> (a) One variant utilizes the length of non-expert trajectories for $l$;
> (b) Another variant estimates the shortest path length heuristically using the L1 distance between the start  $(x_s,y_s)$ and the goal $(x_g, y_g)$, i.e., $D=|x_s-x_g|+|y_s-y_g|$.
>
>
> As indicated in the table below, both overestimation and underestimation led to a performance degradation of no more than 7%. Additionally, we found that leveraging non-expert data or the heuristic L1 distance only yields a nearly 3% degradation in performance, and performs better than the case when the optimal length is extremely overestimated/underestimated. These results imply that employing the information from non-expert data or heuristic estimation could be taken as an alternative when the optimal length is not available.
>
>
> For the details in ViZDoom environment, please refer to our reply to Q3 below.
>
> Table R1: The ablation study for using various estimated lengths of optimal paths for adaptive highway loss, under $35 \times 35 $ ViZDoom navigation.
> The best results are highlighted by **bold**, while the second-best result is highlighted by $\underline{\text{underline}}$.
>
>
>
>
>
>
>
> | Shortest Path Length                                         | [1,100]                     | [100, 200]                  | [200,300]                   |
> | ------------------------------------------------------------ | --------------------------- | --------------------------- | --------------------------- |
> | $\widehat{l}=0$ (connected to all hidden layers)             | $99.49_{\pm 0.35}$          | $94.51_{\pm 0.77}$          | $89.1_{\pm 3.56}$           |
> | $\widehat{l}=l/2$                                            | $99.62_{\pm 0.91}$          | $96.21_{\pm 0.44}$          | $91.24_{\pm 1.68}$          |
> | ${\widehat{l}=l}$                                            | $\mathbf{99.67_{\pm 0.22}}$  | $\mathbf{97.92_{\pm 0.11}}$ | $\mathbf{96.41_{\pm 0.37}}$ |
> | $\widehat{l}=2*l$                                            | $99.61_{\pm 0.18}$          | $96.29_{\pm 0.48}$          | $93.12_{\pm 0.73}$          |
> | $\widehat{l}=N$ (connected to only last layer)               | $99.52_{\pm 0.29}$          | $95.52_{\pm 0.86}$          | $91.12_{\pm 1.64}$          |
> | $\widehat{l}=l \cdot \max(\epsilon,0), \epsilon \sim \mathcal{N}(1,1)$ | $99.62_{\pm 0.50}$          | $96.19_{\pm 0.15}$          | $93.21_{\pm 0.92}$          |
> | $\widehat{l}=len($non-expert path$)$                         | ${99.62_{\pm 0.12}}$        | $\underline{97.01_{\pm 0.69}}$        | ${93.31_{\pm 0.31}}$        |
> | $\widehat{l}=D$ (L1 distance)                            | $\underline{99.64_{\pm 0.49}}$ | ${96.92_{\pm 0.05}}$ | $\underline{93.52_{\pm 0.87}}$ |

---

> ### Author Response · Authors · 2024-11-24
> **Response to Reviewer FzU7 (3/4)**
>
> ---
>  ### W3:
> >  Further, the number of layers has a big impact on the loss scales. Consider the extreme case of 5000 layers, if the optimal path is 1000, then the final loss will have 4000+1 terms, which can make the optimization hard as the effective backpropagation of gradients can be challenging. The paper should discuss how this long-range backpropagation of gradients can be well handled in practice.
>
> ### Answer:
> Thanks for this very insightful comments.
> This is discussed in the appendix. Specifically, Appendix B.1, Line 728,
>
> "To reduce the computational complexity of highway loss, we apply adaptive highway loss only to layers n satisfying the condition $n \mod J = 0$, where $J$ is a hyperparameter set to $10$ in our experiments."
>
> Here, the main idea is to build the highway connections at interval $J$, for example, every 10 neural network layers.
> Using this, the number of the loss terms will reduce to only $1/J$ of the original one.
> As demonstrated in the table below, this mechanism with
> $J=10$ reduces training time to 1/3, while incurring a negligible performance degradation of just 0.01% compared to the configuration without it.
>
>
>
> Table R2: Training Time and Success Rates (%) across Different Ranges of SPLs for DT-VIN with Different $J$ Values
>
>
>
>
> |                                   | Training Time (hours) |                              | Success Rate (%)                 |                             |
> | --------------------------------- | ------------- | ---------------------------- | --------------------------- | --------------------------- |
> |                                   |               | [1,100]                      | [100,200]                   | [200,300]                   |
> | $J=1$ | 37            | $\mathbf{100.00_{\pm 0.00}}$ | $\mathbf{99.99_{\pm 0.01}}$ | $\mathbf{99.78_{\pm 0.21}}$ |
> | $J=10$                            | 12.1          | $\mathbf{100.00_{\pm 0.00}}$ | $\mathbf{99.99_{\pm 0.01}}$ | $99.77_{\pm 0.23}$          |
> | $J=50$                            | 7.1           | $100.00_{\pm 0.00}$          | $99.98_{\pm 0.02}$          | $99.69_{\pm 0.27}$          |
>
>
>
>
>
>
> ## Questions
>
> ---
>  ### Q1:
> >  Since the training of highway connections requires the expert path from the start to the goal, how the training and testing mazes are constructed in the empirical evaluation? If the training and testing mazes are the same, the extra loss might leak the path info to the planning module.
>
> ### Answer:
> Thanks for this thoughtful comment.
> We would like to clarify that the highway loss is used only in the training phase, and not in the testing phase; after training, the learned model directly selects actions without again looking at the expert data.
>
> In accordance with the experimental setup in the GPPN paper, the training and testing mazes are randomly generated using Depth-First Search with the Recursive Backtracker algorithm. We also enforce that all the maze maps, i.e., the positions of the walls and goals, in the dataset are different.

---

> > ### Comment · Reviewer_FzU7 · 2024-11-27
> > **Thanks for the response**
> >
> > Re "all the maze maps, i.e., the positions of the walls and goals, in the dataset are different", do you mean that all mazes in training dataset are different from each other? Are there any mazes that co-exist both in Training and Test datasets? My question is the overlap of same mazes in training and test datasets.

---

> ### Author Response · Authors · 2024-11-24
> **Response to Reviewer FzU7 (4/4)**
>
> ---
>  ### Q2:
> >  In Figure 9, there is no plot of 100x100 maze. Why is such plot not provided? Given that the depth of 5000 is required only for such maze, it would be crucial to see how the choice of other depth values would affect the SR and OR.
>
> ### Answer:
>  Thanks for this useful suggestion.
> We show the performance for $100 \times 100$ maze in Fig. 9 and Fig. 10 in the revised paper. For the convenience of the reviewer, we listed the performance in the table below. For tasks requiring more planning steps, it is crucial to use a network of depth 5000 to solve the task.
>
>
> Table R3: The success rate for DT-VIN across various depths $N$.
>
> | Shortest Path Length | [1,600]                    | [600, 1200]                | [1200,1800]                |
> | -------------------- | -------------------------- | -------------------------- | -------------------------- |
> | $N=600$              | $93.39_{\pm 0.22}$         | $0.32_{\pm 0.03}$          | $0.00_{\pm 0.00}$          |
> | $N=5000$             | $\mathbf{99.98_{\pm0.00}}$ | $\mathbf{99.56_{\pm0.20}}$ | $\mathbf{88.65_{\pm4.76}}$ |
>
>
>
>
>
> Table R4: The optimality rate for DT-VIN across various depths $N$.
>
> | Shortest Path Length | [1,600]                    | [600, 1200]                | [1200,1800]                |
> | -------------------- | -------------------------- | -------------------------- | -------------------------- |
> | $N=600$              | $93.34_{\pm 0.22}$         | $0.28_{\pm 0.03}$          | $0.00_{\pm 0.00}$          |
> | $N=5000$             | $\mathbf{99.98_{\pm0.00}}$ | $\mathbf{99.55_{\pm0.21}}$ | $\mathbf{88.62_{\pm4.78}}$ |
>
>
>
>
> ---
>  ### Q3:
> > The description to ViZDoom experiment is vague and more details are needed. Can the authors clarify how the optimal path can be found and what it means for “3D ViZDoom mazes with size M = 35”?
>
> ### Answer:
>
> In 3D ViZDoom, we follow the setting in the GPPN paper [r1], where the expert data are computed using Dijkstra’s algorithm on the underlying 2D maze map of the environment.
> For "3D ViZDoom with size  $M = 35$ ", we refer to a maze designed on a grid of  $35 \times 35$  cells. Each cell in this grid corresponds to an area of  $64 \times 64 $ map units within the 3D ViZDoom environment. The map unit is the basic measure of space used in the ViZDoom game engine to define distances and sizes. We have revised our paper to clarify this point.
>
> [r1] Lee L et al. Gated path planning networks[C]//ICML, 2018.
>
>
>
>
>
>
>
>
>
> ---
>  ### Q4:
> >  The evaluation on Rover Navigation is also unclear. The paper says that “elevation data are utilized solely for creating expert paths and assessing algorithm performance, not as input for the neural networks”. But the expert paths are needed to define the training loss (i.e., the additional loss), right? Then the proposed method DT-VIN still (implicitly) requires the evaluation data as the input?
>
> ### Answer:
> You're correct that in our experiments, elevation data is implicitly used in training.
> However, this is true for all the baselines, including our proposed DT-VIN and compared method, which use expert data during training, and, like them, DT-VIN does not use expert data during testing.
> Furthermore, even in the training phase, the elevation data is not strictly necessary for DT-VIN---only expert trajectories are necessary (a weaker requirement than methods needing the full dynamics inferred from elevation data, such as A*, would require).
> It's a caveat of this benchmark that we have access to elevation data and can use this to generate expert trajectories. Other methods like Monty Carlo exist to generate expert data without relying on such information and in line with that many benchmarks only provide expert trajectories. For example, the Mars Rover 2020 [r2] benchmark only provides expert trajectories built with programs written by expert rover drivers (we opted against this benchmark as it was not open source).
> We have clarified this point in the revised paper.
>
>
> [r2] Ono, Masahiro, et al. "Data-driven surface traversability analysis for Mars 2020 landing site selection." 2016 IEEE Aerospace Conference. IEEE, 2016.

---

> > ### Comment · Reviewer_FzU7 · 2024-11-27
> > **VizDoom Dijkstra and Evaluation data for Rover Navigation**
> >
> > Re the Dijkstra to find optimal path for ViZDoom and the evaluation data use for Rover Navigation, I'm bit confused: if such groundtruth data have been used or already existed in the training and the test environment remains the same, what's point of using the proposed method to solve such long-term planning? as this "long-term planning" has been solved already and there is no test of the generalizability of the proposed method.

---

> > > ### Author Response · Authors · 2024-11-29
> > > **Further Response to the Reviewer FzU7 (1/3)**
> > >
> > > We would like to start by thanking the Reviewer FzU7 for their time and valuable feedback. We've prepared a response to each of your questions and concerns below.

---

> > > ### Author Response · Authors · 2024-11-29
> > > **Further Response to the Reviewer FzU7 (3/3)**
> > >
> > > ---
> > >  ### Q6:
> > > > Second, re the architecture change from fully connected to convolutional layers, isn't it also coming from the VIN paper? as "CNN-based VIN uses an invariant latent transition kernel"?
> > >
> > > ### Answer:
> > >
> > >
> > >
> > >
> > >
> > >
> > > This is a minor misunderstanding. In the previous answer we wrote:
> > >
> > > > Moreover, the original VIN paper did not specify whether one should employ fully connected or convolutional layers for the transition mapping.
> > >
> > > By "convolutional layers for the transition mapping" we mean that the architecture of the transition mapping module utilizes a CNN architecture. Here, we are not talking about the architecture of the *VI module (planning module)* of VIN, which also uses a CNN architecture (in our paper and in the original VIN paper).
> > >
> > > Although the original VIN proposes a general framework for using an observation-dependent transition kernel in their Method section (Section 3), they did not define the architecture of the transition mapping module---that is, whether it should use fully-connected layers or CNN layers.
> > > This is because, in their experiments, the transition kernel is not only invariant with respect to each latent state (as previously mentioned) but also does not depend on the observation (refer to footnote 3 in Section 4 of the original VIN paper); their transition kernel consists just of learnable parameters.
> > >
> > > Since we are the first to implement an observation-dependent transition kernel, we needed to specify the architecture of such a mapping function.
> > > This is just a minor contribution.
> > > Our major contributions are the dynamic transition kernel and using a highway loss.
> > >
> > >
> > >
> > >
> > >
> > >
> > >
> > >
> > >
> > >
> > >
> > >
> > >
> > >
> > >
> > >
> > >
> > > ---
> > >  ### Q7:
> > > > Re "all the maze maps, i.e., the positions of the walls and goals, in the dataset are different", do you mean that all mazes in training dataset are different from each other? Are there any mazes that co-exist both in Training and Test datasets? My question is the overlap of same mazes in training and test datasets.
> > >
> > > ### Answer:
> > > We apologize for not being clear in our last response.
> > > We ensure that no mazes are shared by the training and test datasets.
> > > Specifically, while each unique arrangement of walls may have various starting positions and goals, we ensure that any given wall arrangement appears exclusively in either the training set or the test set.
> > > This approach prevents any information leakage from the training set to the test set.
> > >
> > >
> > >
> > > ---
> > >  ### Q8:
> > > > Re the Dijkstra to find optimal path for ViZDoom and the evaluation data use for Rover Navigation, I'm bit confused: if such groundtruth data have been used or already existed in the training and the test environment remains the same, what's point of using the proposed method to solve such long-term planning? as this "long-term planning" has been solved already and there is no test of the generalizability of the proposed method.
> > >
> > > ### Answer:
> > >
> > >
> > >
> > >
> > >
> > >
> > > As previously mentioned, the training and test datasets are different in our experiments.
> > > During testing, we do not use any ground-truth data when running either our method or the baselines (e.g., we don't look at the elevation data or any expert trajectories, which might not be available or could be (like elevation data) expensive to produce); instead, we explicitly evaluate their generalization capabilities in environments wholly not encountered during training.
> > >
> > > At training time, we use Dijkstra's algorithm to produce some expert trajectories (which may not be available at test time).
> > > In other settings, expert trajectories may also just be generally available or produced with a different method like Monte Carlo etc.
> > > At test time Dijkstra's algorithm is used solely to calculate the Optimality Rate of the algorithms for external evaluation purposes (i.e., only to draw the figures in the paper).
> > > Dijkstra's algorithm only works here because we give it access to "illegal data" (elevation data/true ViZDoom 2D map) that would not be available at test time in upstream tasks.
> > > Calculating the optimality rate is important for assessing the general planning capabilities of various methods. This practice is analogous to using labeled data in a test set to evaluate a predictive model in supervised learning.
> > > Note that we would not need to use Dijkstra's at test time if we were reporting only the results in terms of Success Rate for the methods.
> > >
> > >
> > > ---
> > > We again thank the reviewer for their time and work so far. We believe their input has helped greatly strengthen this paper.

---

> ### Author Response · Authors · 2024-11-29
> **Further Response to the Reviewer FzU7 (2/3)**
>
> ---
>  ### Q5:
> > Concerns about the novelty (observations-depdent, CNN).
>
>
>
>
>
> ### Answer:
>
> We think there has been a bit of confusion here and we apologize for not being more clear in our last response.
> Our proposed "*dynamic transition kernel*" goes well beyond the the "*observation-dependent transition kernels*" that are partly described in VIN.
> While observation-dependent transition kernels and the dynamic transition kernels we propose here both depend on the observation $x$ (formally, $\overline{\mathsf{T}}=f^{\overline{\mathsf{T}}}(x)$),
> **the main innovation of our approach lies in the fact that our dynamic kernels are also a function of the latent state.
> In contrast, the observation-dependent kernel of VIN are not.**
>
>
>
> Formally, this means that (as shown in eq. (1) of the paper and reproduced below), the latent transition kernel ${\overline{\mathsf{T}}}_{i,j,\overline{a},i^{\prime},j^{\prime}}$ is different for each latent state $(i,j)$:
>
>
>
> $$\overline{V}\_{i,j}^{(n)} = \max\_{\overline{a}} \sum\_{i',j'} \overline{\mathsf{T}}\_{i,j,\overline{a},i',j'}^{\text{dyn}}
> \left( \overline{\mathsf{R}}\_{i-i',j-j'} + \overline{V}\_{i-i',j-j'}^{(n-1)} \right)$$
>
>
>
>
> **This novel dynamic formulation for the kernel greatly improves the representational capacity of the kernel and is never proposed or implemented in the original VIN nor---to the best of our knowledge---in any later studies.**
> In VIN, the transition kernel remains invariant across different latent states, meaning the same transition kernel ${\overline{\mathsf{T}}}_{\overline{a},i^{\prime},j^{\prime}}$ is used for every different latent state $(i,j)$.
>
> [
> Note that in the [VIN paper](https://proceedings.neurips.cc/paper/2016/file/c21002f464c5fc5bee3b98ced83963b8-Paper.pdf), they use $W_{l, i, j}^{\overline{a}}$ is used instead of ${\overline{\mathsf{T}}}_{\overline{a},i^{\prime},j^{\prime}}$; see the second paragraph of Sec. 3.1 in the original the [VIN paper](https://proceedings.neurips.cc/paper/2016/file/c21002f464c5fc5bee3b98ced83963b8-Paper.pdf).
> ]
>
>
>
>
>
>
>
>
>
>
> We summarize this again below.
> The transition kernels can be additionally dependent on **(1) the observation**, and **(2) the latent state**.
> In VIN, they propose transition kernels that are dependent on **(1) the observation** but they then formalize kernels that are independent of **(1) the observation**.
> All their kernels are independent of **(2) the latent state**.
> We propose kernels that are dependent on both **(1) the observation**, and **(2) the latent state**.
> The key innovation here is the dynamic nature of our kernels that comes from **(2) the latent state**.
> The below table summarizes the differences between the kernels.
>
>
>
>
> Table R4: A comparison of different latent transition kernels.
>
>
>
> | Types of Kernels                  | Comments                                   | dependent on observation? | dependent on the latent state? |
> | --------------------------------- | ------------------------------------------ | :-------------------: | :---------------------: |
> | invariant transition kernel       | proposed and implemented in VIN                         |          NO           |           NO            |
> | observation-dependent kernel      | partly proposed in VIN paper, but not implemented |          **YES**          |           NO            |
> | latent state-dependent kernel      | used here for ablation study |          NO          |           **YES**            |
> | dynamic transition kernel  (ours) | proposed and implemented in our paper                      |          **YES**          |           **YES**           |
>
>
>
> To further demonstrate the importance of this difference, we've run an ablation study to compare the four variants (i.e., independent of (1) and (2), dependent on (1) but not (2), dependent on (2) but not (1), and dependent on both (1) and (2)). The results are shown below. Here the importance of this surgical difference is apparent, with removing dependency on either (1) or (2) leading to a rapid collapse.
>
>
>
>
>
>
>
> Table R5: The success rate for each method in 35×35 2D maze navigation.
>
>
> | Shortest Path Length              | [1,150]                      | [150,300]                    | [300,500]                   |
> | --------------------------------- | ---------------------------- | ---------------------------- | --------------------------- |
> | invariant transition kernel       | $68.44_{\pm 3.12}$           | $0.03_{\pm 0.01}$            | $0.00_{\pm 0.00}$           |
> | observation-dependent kernel      | $71.44_{\pm 4.31}$           | $0.09_{\pm 0.02}$            | $0.00_{\pm 0.00}$           |
> | latent state-dependent kernel     | $73.44_{\pm 3.09}$           | $1.44_{\pm 0.37}$            | $0.00_{\pm 0.00}$           |
> | dynamic transiiton kernel  (ours) | $\mathbf{100.00_{\pm 0.00}}$ | $\mathbf{99.99_{\pm 0.01}}$ | $\mathbf{99.77_{\pm 0.23}}$ |

---

> ### Author Response · Authors · 2024-12-02
> **Discussion Period Ending Soon**
>
> Dear reviewer FzU7, as the discussion period is nearing its end, please let us know if our recent rebuttal has successfully put to rest your concerns. We have made extensive efforts to address both your original and new comments, revising the paper as needed (in accordance with the expectations of the ICLR reviewing process). We understand you likely have many pressing responsibilities, and we appreciate the time you've already dedicated to reviewing our work. We are ready to respond if there are any remaining questions. Thank you once again for your valuable feedback.

---

### Official Review · Reviewer_Pndq · 2024-10-29

**Soundness:** 3
**Presentation:** 4
**Contribution:** 3
**Rating:** 6
**Confidence:** 3

**Summary:**

This paper addresses the topic of long-term large-scale planning in Reinforcement Learning environments, by building on the Value Iteration Network. The authors identify two key deficiencies in VIN, and propose methods to overcome both: 1) the use of a dynamic transition kernel to improve the VIN’s representational capacity, and 2) the use of an “adaptive highway loss” to overcome problems with vanishing or exploding gradients, thus allowing the model to scale to a much greater depth. This deeper, more expressive model is able to solve problems at a scale which defeated the original VIN approach, and its more recent variants.

**Strengths:**

Overall, this paper is well presented, and clearly describes a step forward in the development of VIN-based RL planning. The deficiencies of the original VIN approach are well explained and demonstrated empirically, providing clear motivations for the modifications the authors suggest.
In terms of novelty, the two key innovations (the transition mapping module, and the adaptive highway loss) seem like neat iterations to the previous work, rather than being ground-breaking new directions, but I believe the authors demonstrate that there is real value to them. In particular, it is pleasing to see how their highway loss both improves on and simplifies previous work.

This paper could be a solid step forward, except for some concerns I have regarding motivation and evaluation, discussed below.

**Weaknesses:**

At the detailed level, the motivation is well constructed: this paper improves on several deficiencies in the established approaches. What I feel the paper lacks is a motivation from a broader perspective: why do we need an RL agent that can solve bigger mazes? Particularly when – as acknowledged by the authors – the A* algorithm has been around since 1968 (and, indeed, Dijkstra’s algorithm is used in this work to compute the shortest path lengths).

This is linked to my concerns about evaluation: all three domains (2D grid-world mazes, the 3D VizDoom, and the Rover navigation) feel like essentially the same problem. In particular, I can’t really see what value the VizDoom experiments add. As per the GPPN paper, “The maze design for the 3D mazes are generated in exactly the same manner as the 2D mazes…” – this is essentially the same environment, just reconstructed in VizDoom, but with the additional step that a network then has to translate the first-person images back into a 2D top-down representation which is then fed into the DT-VIN – with all the same hyperparameters and settings as the first domain. While I understand the value of using the same benchmarks as the previous papers, the introduction of VizDoom as a benchmark in the first place feels misleading. The suggestion is of an agent playing Doom; in reality, artificially constructing a 2D maze in the VizDoom environment, then taking a full set of N/S/E/W first-person images from each location, then turning these back into a fully-observed 2D representation, is about as far removed from the actual game dynamics as it is possible to get. Perhaps I am missing the point, but beyond training the network to reconstruct the images into the 2D map (which is not novel), this environment doesn’t demonstrate any new capabilities in the DT-VIN, and feels like a wasted opportunity. Instead, I would have loved to see how DT-VIN performed in a very different domain. For example, the original VIN paper evaluated the model in discrete and continuous path-planning, and a natural-language-based search domain. As mentioned in this paper, “VINs have been shown to perform well in… path planning, autonomous navigation, and complex decision-making in dynamic environments”, but DT-VIN was only evaluated in a very narrow set of domains.

As the introduction put it, “Planning is the problem of finding a sequence of actions that achieve a specific pre-defined goal.” – with this broad definition of planning, it is easy to see the motivation for an agent that improves on previous approaches, but the paper’s narrow focus on the already-solved problem of path-planning obscured this motivation.

I’d like to see the authors address these concerns about motivation and evaluation as a first priority.

**Questions:**

Some lower priority questions I had:
1. The formulation of the highway loss requires knowledge of the shortest path l (as acknowledged by the authors in Appendix A) – where this is unknown, what is the cost of over/underestimating it? It would be interesting to see what effect this has on training if it is set too low, or too high.
2. In the GPPN paper, three transition kernels were used (NEWS/Moore/Differential Drive) – were these considered for DT-VIN? I assume NEWS was used in this paper, but I can’t seem to find it.
3. I have some questions about scaling. The authors highlight the “potential of our method to scale to increasingly complex planning tasks alongside the increasing availability of computing power” but I’m uncomfortable with the twin assumptions that compute will keep getting cheaper, and that the method can therefore keep scaling up. I’d like to see some consideration given to these aspects of scale:
  a. _Compute_: GPU training times are only given for one datapoint (the 35x35 maze, with a 600 layer network). How much GPU time was needed to train the 100x100 maze, 5000 layers?
  b. _Model size_: What is the relationship between the number of planning steps required by the problem, and the depth of the network required to solve it? From the plots it appears that success tails off as the number of steps approaches the network depth, but is it feasible to keep increasing network depth as the problems become more complicated?
  c. _Data_: Is it necessary to keep scaling up the dataset in line with increasing the model depth, and does this limit the method’s applicability to other problem domains where training data can’t be so easily generated?

Overall, this was a well-presented and interesting paper. I look forward to being persuaded that my misgivings about motivation and evaluation are unfounded!

---

> ### Author Response · Authors · 2024-11-24
> **Response to Reviewer Pndq (1/6)**
>
> We sincerely appreciate Reviewer Pndq's constructive comments.
> Below we address the questions and concerns.
>
> ## Weaknesses
>
>
> ---
>  ### W1:
> > What I feel the paper lacks is a motivation from a broader perspective: why do we need an RL agent that can solve bigger mazes? Particularly when – as acknowledged by the authors – the A* algorithm has been around since 1968 (and, indeed, Dijkstra’s algorithm is used in this work to compute the shortest path lengths).......
>
> > This is linked to my concerns about evaluation: all three domains (2D grid-world mazes, the 3D VizDoom, and the Rover navigation) feel like essentially the same problem.
> In particular, I can’t really see what value the VizDoom experiments add. ......
>
>
> ### Answer:
>
> Thank you for your comments regarding the motivation and evaluation in our paper.
>
> First, we would like to note that traditional search-based algorithms such as A* and Dijkstra’s algorithm require a fully known and accurate environmental model to function effectively. This condition limits their applicability in scenarios where the MDP transitions or rewards are unknown or the state and action spaces are large or continuous. Our work instead considers environments without an accurate model, focusing on learning to generalize across unseen mazes directly from visual inputs rather than predefined graph structures.
> For example, in continuous control problems where the state and action space are continuous (as we will show in response to W2), classical search-based planning algorithms like A*, are ill-suited. In contrast, our model learns directly from observations to actions, demonstrating generalizability to unseen tasks.
>
>
>
> While our tasks are fundamentally based on navigation and planning, they address distinct challenges that collectively showcase the broad applicability and robustness of our proposed method in both simulated and real-world scenarios.
>
> The 3D ViZDoom environment (Sec 4.2) was selected to assess the planning abilities of our algorithms under conditions of perceptual uncertainty, where the input is derived from noisy, first-person visual inputs. This introduces a significantly different challenge that tests the algorithm’s effectiveness in environments similar to those encountered in robotics.
> The sentence "The maze designs for the 3D mazes are generated in exactly the same manner as the 2D mazes…" in GPPN just refers to the way the positions of obstacles are generated. As also mentioned by the reviewer, the input to the model of 3D ViZDoom is different from the 2D maze, and the model needs to predict a 2D maze, which could be noisy and bring difficulty to the planning module.
>
>
> The Rover navigation task involves long-distance planning based on satellite imagery, presenting a practical challenge where short-term planning solutions are inadequate. This task is especially demanding due to complex real-world details such as variable lighting, textures, and the complications introduced by aerial photography.

---

> > ### Comment · Reviewer_Pndq · 2024-11-25
> >
> > Thank you for your detailed responses. I recognise that "Can't we already do this with A*?" is a tedious question for RL researchers, so thank you for replying graciously!
> >
> > >  Our work instead considers environments without an accurate model, focusing on learning to generalize across unseen mazes directly from visual inputs rather than predefined graph structures.
> >
> > Unless I've misunderstood, the first environment you evaluate under *does* have an accurate model - isn't the agent given a fully observed map of the maze? This *is* arguably a predefined graph structure - or, at least, it's a very small step to turn a grid of pixels into a graph.
> >
> > I understand that DT-VIN is intended to perform in situations where classical planning approaches would fail; unfortunately, this first environment seems to be a case which is ideally suited to classical planning approaches, and as such I don't feel like it really showcases the advantages of DT-VIN.
> >
> > > The 3D ViZDoom environment (Sec 4.2) was selected to assess the planning abilities of our algorithms under conditions of perceptual uncertainty, where the input is derived from noisy, first-person visual inputs.
> >
> > This is probably an unfair criticism, given that this approach already seems to have been enshrined as a baseline by prior papers, but it feels very Rube-Goldberg... to check my understanding: a 2D maze is generated, then turned into a "3D" Doom map, then screenshots are taken at each position and cardinal orientation, and then these images are fed into a pretrained(?) model which has been trained to turn this _back_ into a 2D maze. *This* 2D maze is then given to the planning module _in the same way_ as in the first environment? Am I right in thinking that this process (2D to 3D then back to 2D) all takes place _apart_ from DT-VIN? i.e from DT-VIN's point of view, the details of this process are irrelevant - it's now just seeing a slightly less accurate version of the same sort of maze it was tested on in the first experiment? (Apologies if I've misunderstood this.)
> >
> > If my understanding is correct, the presence of Doom here still feels like a red-herring, from a planning perspective. I feel like a better experiment would be to take the original maze environment, and then corrupt the mazes in some more direct, straight-forward, controllable way. This would surely give a clearer insight into the performance of DT-VIN, without all the Doom-laden obfuscation.
> >
> > > The Rover navigation task involves long-distance planning based on satellite imagery, presenting a practical challenge where short-term planning solutions are inadequate. This task is especially demanding due to complex real-world details such as variable lighting, textures, and the complications introduced by aerial photography.
> >
> > Yes, and re-reading now, I see the interesting comparison with CNN+A*. The new continuous environment is also interesting.

---

> ### Author Response · Authors · 2024-11-24
> **Response to Reviewer Pndq (2/6)**
>
> ---
>  ### W2:
> > I would have loved to see how DT-VIN performed in a very different domain. .....
>
> ### Answer:
>
> In response to your suggestion for diversity in our experimental domains, we have extended our evaluation to a *continuous control setting*, where both state and action spaces are continuous. Here, we use the Point Maze as the benchmark [r1], where the agent must decide the forces to apply to a controllable ball to reach a goal within 800 steps. We measure both the final distance to the goal and the success rate, where we say an agent has succeeded if the Euclidean distance to the goal is less than 0.5.
> As shown in the table below, DT-VIN achieved closer proximity to the goal and a higher success rate compared to the other methods.
>
>
>
>
>
> Table R2: The performance of each method on continuous control tasks.
>
>
>
> |               | Final Distance to the Goal (Euclidean Distance) | Success Rate (%)           |
> | :-----------: | :----------------------------------------: | --------------------------- |
> |      VIN      |            ${5.12_{\pm 3.19}}$             | ${62.00_{\pm 2.18}}$        |
> |     GPPN      |            ${4.12_{\pm 2.18}}$             | ${68.12_{\pm 4.17}}$        |
> |  Higwhay VIN  |             $4.98_{\pm 3.28}$              | ${67.31_{\pm 3.28}}$        |
> | DT-VIN (ours) |          $\mathbf{2.28 \pm 1.2 }$          | $\mathbf{82.00_{\pm 3.89}}$ |
>
>
>
>
> [r1] Fu, Justin, et al. "D4RL: Datasets for Deep Data-Driven Reinforcement Learning." arXiv preprint arXiv:2004.07219 (2020).

---

> > ### Comment · Reviewer_Pndq · 2024-11-25
> >
> > Thanks, I think this is a great addition.

---

> ### Author Response · Authors · 2024-11-24
> **Response to Reviewer Pndq (3/6)**
>
> ## Questions
>
>
>
>
>
> ---
>  ### Q1:
> > The formulation of the highway loss requires knowledge of the shortest path $l$ (as acknowledged by the authors in Appendix A) – where this is unknown, what is the cost of over/underestimating it? ...
>
> ### Answer:
>
>
> Thanks for this insightful comment.
> Note that the knowledge of the length $l$ of the expert path naturally exists in the imitation learning case.
> However, for the case where such information is unknown, one can use either the length of non-expert data or some heuristic methods to estimate $l$ when the actual $l$ is completely unknown, e.g., using the distance between the start and the goal position.
> To further address your comment, we have now run a number of ablations on this.
>
>
>
>
> First, to measure the effect of overestimation/underestimation, we conducted the experiment with various estimated values of the length of the shortest path $\widehat{l}$, which are $0, l/2, l, 2l, N$ (where $l$ is the actual length of the shortest path, $N$ is the depth of the planning module).
> Second, to evaluate the case when the etsimation of $l$ has variance, we use $l \cdot \max(\epsilon, 0)$ as the estimation, with $\epsilon$ sampled from a Gaussian distribution $\mathcal{N}(1, 1)$.
> Third, we also assessed two additional variants for estimating $l$:
> (a) One variant utilizes the length of non-expert trajectories for $l$;
> (b) Another variant estimates the shortest path length heuristically using the L1 distance between the start  $(x_s,y_s)$ and the goal $(x_g, y_g)$, i.e., $D=|x_s-x_g|+|y_s-y_g|$.
>
>
> As indicated in the table below, both overestimation and underestimation led to a performance degradation of no more than 7%. Additionally, we found that leveraging non-expert data or the heuristic L1 distance only yields a nearly 3% degradation in performance, and performs better than the case when the optimal length is extremely overestimated/underestimated. These results imply that employing the information from non-expert data or heuristic estimation could be taken as an alternative when the optimal length is not available.
>
>
> Table RX: The ablation study for using various estimated lengths of optimal paths for adaptive highway loss, under $35 \times 35 $ ViZDoom navigation.
> The best results are highlighted by **bold**, while the second-best result is highlighted by $\underline{\text{underline}}$.
>
>
>
>
>
>
>
> | Shortest Path Length                                         | [1,100]                     | [100, 200]                  | [200,300]                   |
> | ------------------------------------------------------------ | --------------------------- | --------------------------- | --------------------------- |
> | $\widehat{l}=0$ (connected to all hidden layers)             | $99.49_{\pm 0.35}$          | $94.51_{\pm 0.77}$          | $89.1_{\pm 3.56}$           |
> | $\widehat{l}=l/2$                                            | $99.62_{\pm 0.91}$          | $96.21_{\pm 0.44}$          | $91.24_{\pm 1.68}$          |
> | ${\widehat{l}=l}$                                            | $\mathbf{99.67_{\pm 0.22}}$  | $\mathbf{97.92_{\pm 0.11}}$ | $\mathbf{96.41_{\pm 0.37}}$ |
> | $\widehat{l}=2*l$                                            | $99.61_{\pm 0.18}$          | $96.29_{\pm 0.48}$          | $93.12_{\pm 0.73}$          |
> | $\widehat{l}=N$ (connected to only last layer)               | $99.52_{\pm 0.29}$          | $95.52_{\pm 0.86}$          | $91.12_{\pm 1.64}$          |
> | $\widehat{l}=l \cdot \max(\epsilon,0), \epsilon \sim \mathcal{N}(1,1)$ | $99.62_{\pm 0.50}$          | $96.19_{\pm 0.15}$          | $93.21_{\pm 0.92}$          |
> | $\widehat{l}=len($non-expert path$)$                         | ${99.62_{\pm 0.12}}$        | $\underline{97.01_{\pm 0.69}}$        | ${93.31_{\pm 0.31}}$        |
> | $\widehat{l}=D$ (L1 distance)                            | $\underline{99.64_{\pm 0.49}}$ | ${96.92_{\pm 0.05}}$ | $\underline{93.52_{\pm 0.87}}$ |

---

> > ### Comment · Reviewer_Pndq · 2024-11-26
> >
> > These ablations on the length `l` are a great addition to the paper. If I'm understanding correctly, even at the worst case when `l` is set to 0 for the [200,300] shortest path length range, DT-VIN is still significantly outperforming the other baselines (for the 35x35 VizDoom experiments, according to fig. 6b).
> > This seems like a fairly comprehensive answer to the concern about the estimation for `l` being a potential weakness in the method. Are similarly strong results seen in the other environments, eg continuous control? (Not asking for new experiments to be run or added to the paper, just curious if you'd tried this and had any anecdotal evidence?)

---

> ### Author Response · Authors · 2024-11-24
> **Response to Reviewer Pndq (4/6)**
>
> ---
>  ### Q2:
> >  In the GPPN paper, three transition kernels were used (NEWS/Moore/Differential Drive) – were these considered for DT-VIN? ......
>
> ### Answer:
> Yes, we used the NEWS transition kernel for the maze environment because it's the standard one used in gridworld and maze environments, and our preliminary experiments didn't find a notable difference when swapping to different transition kernels.
> We mentioned this setting in Lines 269-270 in the original paper, the agent navigates the four adjacent cells by moving one step at a time in any of the four cardinal directions.
> We have run an additional ablation using different transition kernels. As shown in the tables below, DT-VIN consistently outperforms all the compared methods regardless of the kernel used.
>
>
>
>
>
>
> Table R5: The success rate for each method in $35\times 35$ 2D maze navigation with **Differential Drive** transition kernel, where the agent can move forward along its orientation or rotate 90 degrees left or right.
>
> | Shortest Path Length | [1,150]                      | [150,300]                    | [300,500]                   |
> | -------------------- | ---------------------------- | ---------------------------- | --------------------------- |
> | VIN                  | $68.44_{\pm 3.12}$           | $0.03_{\pm 0.01}$            | $0.00_{\pm 0.00}$           |
> | GPPN                 | $83.1_{\pm 1.23}$            | $0.31_{\pm 0.01}$            | $0.0_{\pm 0.0}$             |
> | Higwhay VIN          | $87.1_{\pm 3.73}$            | $57.1_{\pm 3.98}$            | $49.1_{\pm 8.73}$           |
> | DT-VIN (ours)        | $\mathbf{100.00_{\pm 0.00}}$ | $\mathbf{100.00_{\pm 0.00}}$ | $\mathbf{99.99_{\pm 0.01}}$ |
>
>
>
>
>
>
> Table R6: The success rate for each method in $35\times 35$ 2D maze navigation with **MOORE** transition kernel, where the agent can relocate to any of the eight adjacent cells that comprise its Moore neighborhood.
>
> | Shortest Path Length | [1,100]                     | [100,200]                  | [200,250]                  |
> | -------------------- | --------------------------- | -------------------------- | -------------------------- |
> | VIN                  | $66.44_{\pm 3.21}$          | $0.00_{\pm 0.00}$          | $0.00_{\pm 0.00}$          |
> | GPPN                 | $89.94_{\pm 1.31}$          | $0.04_{\pm 0.01}$          | $0.00_{\pm 0.00}$          |
> | Higwhay VIN          | $83.14_{\pm 2.21}$          | $37.1_{\pm 1.98}$          | $25.1_{\pm 3.28}$          |
> | DT-VIN (ours)        | $\mathbf{100.0_{\pm 0.00}}$ | $\mathbf{98.9_{\pm 0.72}}$ | $\mathbf{96.7_{\pm 1.23}}$ |

---

> > ### Comment · Reviewer_Pndq · 2024-11-25
> >
> > Thank you too for all the detailed extra work on scaling and transition kernels - I'll need to take a little more time to digest all this, and will try to come back with a response as quickly as I can tomorrow.

---

> > ### Comment · Reviewer_Pndq · 2024-11-26
> >
> > (on the different transition kernels)
> >
> > Thanks for this extra work - interesting that the `Moore` kernel performs less well than the `Differential Drive` kernel. I'd have expected the opposite, given the added complexity of the state space and path length required by Differential Drive. Am I right that DT-VIN (and others) score highest on Diff Drive, then NEWS, then Moore? Does this indicate an inverse relationship between size of action space and performance? (Asking mostly for my own interest.)

---

> ### Author Response · Authors · 2024-11-24
> **Response to Reviewer Pndq (5/6)**
>
> ---
>  ### Q3:
> > ... Concerns about scale:
>
> > a. Compute: ... How much GPU time was needed to train the 100x100 maze, 5000 layers?
>
> > b. Model size: What is the relationship between the number of planning steps required by the problem, and the depth of the network required to solve it? ...
> From the plots, it appears that success tails off as the number of steps approaches the network depth, but is it feasible to keep increasing network depth as the problems become more complicated?
>
> > c. Data: Is it necessary to keep scaling up the dataset in line with increasing the model depth, and does this limit the method’s applicability to other problem domains where training data can’t be so easily generated?
>
> ### Answer to Q3 (1/2):
>
>
> **a. Compute:**
> Table R7 shows the memory consumption and training time on an NVIDIA A100 GPU for DT-VIN and the compared methods when using 5000 layers and training for $90$ epochs on $100 \times 100$ maze.
> As shown in the table below, although DT-VIN takes more training time than GPPN, it requires much less GPU memory.
> These results are generally consistent with those observed in the $35 \times 35$ 2D maze in Table 5 of the original paper.
>
> Table R7: Computational complexity of each method using 5000 layers and training for $90$ epochs, evaluated on $100 \times 100$ 2D maze navigation.
>
> | Method        | GPU Memory (GB) | Training Time (hours) |
> | ------------- | :-------------: | :-------------------: |
> | VIN           |       35        |          36           |
> | GPPN          |       710       |          31           |
> | Highway VIN   |       111        |          112           |
> | DT-VIN (ours) |       182       |          98           |
>
>
> **b.Model size:**
> In our experiments, the depth of the network required to solve the problem is close to *linear* with the number of planning steps required by the problem.
> For maze size $M=15, 25, 35$, we test DT-VIN models at increasing depths in increments of 100 until the optimal performance is achieved.
> For instance, for mazes of size $25\times25$, we assess depths of $100,200,300,400$.
> For maze size $M=100$, we assess depths of $4000, 5000, 6000$.
> As Table R8 illustrates, the depth of the smallest network that can solve the task increases slightly more than linearly with the required planning steps.  Therefore, it might be feasible to continue increasing the network depth as the problems become more complex.
>
> Table R8: Minimal depths of DT-VIN model across various maze sizes.
>
> | Maze Size | Longest Length of Optimal Path | Minimal Depth of DT-VIN |
> | --------- | ------------------------------ | ------------- |
> | 15        | 80                             | 100           |
> | 25        | 200                            | 300           |
> | 35        | 300                            | 500           |
> | 100       | 1800                           | 5000          |

---

> > ### Comment · Reviewer_Pndq · 2024-11-26
> >
> > (on scaling the model depth)
> > Thank you, this information is good to see, and I think it makes the original claim (about the “potential of our method to scale to increasingly complex planning tasks alongside the increasing availability of computing power") much less hand-wavy.
> >
> > Small question on Table R7 for my own understanding - where the baselines require, eg, 710GB of GPU memory, how is this achieved on a single A100 (which only has 40 or 80GB VRAM)?

---

> ### Author Response · Authors · 2024-11-24
> **Response to Reviewer Pndq (6/6)**
>
> ### Answer to Q3 (2/2):
>
> **c. Data:**
> The scale of the dataset needs to scale up with the complexity of the problem rather than the model depth.
> Under the same scale of the problem, we didn't find that increasing model depth requires additional data. As shown in Table R9, without expanding the dataset, increasing the model depth does not reduce the performance.
>
>
>
> Table R9: The success rate of DT-VIN across various dataset depths $N$, maintaining the same size as the original dataset.
>
> | Shortest Path Length | [1,100]             | [100,200]          | [200,300]          |
> | -------------------- | ------------------- | ------------------ | ------------------ |
> | $N=300$              | $99.99_{\pm 0.01}$  | $99.81_{\pm 0.13}$ | $92.11_{\pm 1.31}$ |
> | $N=600$              | $100.00_{\pm 0.00}$ | $99.99_{\pm 0.01}$ | $99.77_{\pm 0.23}$ |
> | $N=1200$             | $100.00_{\pm 0.00}$ | $99.99_{\pm 0.01}$ | $99.81_{\pm 0.11}$ |
>
>
> Moreover, even in situations where data is rare, DT-VIN still outperforms compared methods.
> As shown in Table R10, with only 50% of the original dataset, DT-VIN greatly outperforms existing methods.
> We also highlight the changes compared to the performance with a full-sized dataset in Table R11, where DT-VIN results in less than a $0.2%$ degradation for tasks within the range [1, 100], while the best-performing comparison method, GPPN, incurs a degradation of nearly $12%$.
>
> Table R10: The **success rate** for each method, using a dataset reduced to **50%** of the original size.
>
> | Shortest Path Length | [1,100]                     | [100,200]                  | [200,300]                   |
> | -------------------- | --------------------------- | -------------------------- | --------------------------- |
> | VIN                  | $32.41_{\pm 4.25}$          | $0.00_{\pm 0.00}$          | $0.00_{\pm 0.00}$           |
> | GPPN                 | $83.11_{\pm 1.33}$          | $0.01_{\pm 0.01}$          | $0.00_{\pm 0.00}$           |
> | Higwhay VIN          | $45.41_{\pm 4.13}$          | $37.41_{\pm 3.25}$         | $21.41_{\pm 6.98}$          |
> | DT-VIN (ours)        | $\mathbf{99.96_{\pm 0.01}}$ | $\mathbf{99.8_{\pm 0.12}}$ | $\mathbf{96.01_{\pm 0.32}}$ |
>
>
>
>
>
> Table R11: The **changes** in success rate for each method, using a dataset reduced to **50%** of the original size, compared to the full-sized dataset (more negative is worse).
>
> | Shortest Path Length | [1,100]                     | [100, 200]                  | [200, 300]                  |
> | -------------------- | --------------------------- | --------------------------- | --------------------------- |
> | VIN                  | $-36.00_{\pm 3.12}$         | $0.00_{\pm 0.00}$           | $0.00_{\pm  0.00}$          |
> | GPPN                 | $-12.60_{\pm1.29}$          | $-0.38_{\pm 0.11}$          | $0.00_{\pm 0.00}$           |
> | Highway VIN          | $-45.26_{\pm 3.48}$         | $-28.09_{\pm 2.98}$         | $-32.99_{\pm 3.11}$         |
> | DT-VIN (ours)        | $\mathbf{-0.04_{\pm 0.01}}$ | $\mathbf{-0.19_{\pm 0.04}}$ | $\mathbf{-3.76_{\pm 0.31}}$ |

---

> > ### Comment · Reviewer_Pndq · 2024-11-26
> >
> > (on scaling the dataset)
> >
> > Good to see that the data scales with the problem complexity rather than the model size (though presumably you'd ideally aim to keep the model as small as required anyway).
> >
> > Tables R10 and R11 seem to show that DT-VIN is much more data efficient than the other baselines, which is also encouraging to see. This doesn't fully answer the question of *how* the dataset scales with the problem complexity, though. Is this also linear (or close to linear?) (I realise it's probably not possible to answer this empirically in the given timeframe, am happy for this to remain unanswered - unless there is related evidence in other papers that can be cited as an indication? My slight concern is that, as I understand it, you are training using IL from data labelled with known expert solutions. If there's a lurking exponential growth in the required dataset size, as problems scale in complexity, training will end up relying on environments for which expert solutions can be calculated easily (eg simple mazes labelled using A*), and then petty reviewers will be able to make snarky comments about how RL algorithms aren't as useful as classical planning algorithms...) Perhaps this is for a follow-up paper though.
> >
> > This slight concern aside, I think the tables showing DT-VIN's data efficiency make a strong argument for its supremacy over the previous baselines.
> >
> > (There's a small typo in the description here: (table 14 in the paper) "Table R9: The success rate of DT-VIN across various *dataset* depths, maintaining the same size as the original dataset." - should presumably read "model depths".)

---

> ### Comment · Reviewer_Pndq · 2024-11-26
>
> The extra work over the last week or two has, I believe, clearly demonstrated the proposed method's superiority over the previous baselines when it comes to other environments (eg continuous action space), data efficiency, and different transition kernels. The authors have also successfully shown that the method's reliance on knowing in advance the length of the expert path (`l`) isn't the weakness it first appeared to be. In the light of this, I'm revising my assessment of the paper's contribution, and my overall rating.

---

> > ### Author Response · Authors · 2024-11-29
> > **Further Response to the Reviewer Pndq (1/3)**
> >
> > Thank you so much for taking much time and effort to carefully read our response and for increasing your score. We believe that your suggestions and comments have helped us improve the quality of this paper.
> >
> > We are glad that most of our responses have addressed your concerns.
> > Please see below for our responses to your follow-up questions.
> >
> >
> > ---
> >  ### Q4:
> > > Unless I've misunderstood, the first environment you evaluate under does have an accurate model - isn't the agent given a fully observed map of the maze? This is arguably a predefined graph structure - or, at least, it's a very small step to turn a grid of pixels into a graph.
> >
> > > I understand that DT-VIN is intended to perform in situations where classical planning approaches would fail; unfortunately, this first environment seems to be a case which is ideally suited to classical planning approaches, and as such I don't feel like it really showcases the advantages of DT-VIN.
> >
> > ### Answer:
> > Sorry, we should have written, "Our work instead considers **mostly** environments without an accurate model". The 2D maze environment remains useful for directly comparing the planning capabilities of DT-VIN with its predecessors, including VIN, GPPN, and Highway VIN. A notable advantage of using 2D mazes is the straightforward assessment of the planning difficulty of a specific example and the ability to smoothly generate mazes that match a desired level of complexity. We agree that other benchmarks we propose, such as ViZDoom, Rover navigation, and continuous control, are more appropriate for showcasing the advantages of DT-VIN in situations where classical planning approaches are challenging to apply.
> >
> >
> >
> > ---
> >  ### Q5:
> > > Concerns about 3D Maze.
> >
> > > ... I feel like a better experiment would be to take the original maze environment, and then corrupt the mazes in some more direct, straight-forward, controllable way...
> >
> >
> >
> > ### Answer:
> >
> >
> > Your understanding is mostly correct. A network $N$ is trained to map from the set of all the first-person images from one maze to the 2D representation of the maze using a loss $A$. After training $N$ on all the mazes in the training set, we then prepend this to the DT-VIN network $M$ which has the highway loss $B$. We then train the resulting network ($N$ concat $M$; the output of $N$ is fed to $M$) with the loss $A + B$ end-to-end using the first-person views of the mazes in the training set. Next, we directly apply this large network on the unseen test environments. Without the pretraining phase or the additional loss, it would be very computationally expensive to train DT-VIN end-to-end.
> >
> > What we want to determine with ViZDoom, is whether DT-VIN can handle working with a *complex learned* represenation, as one might see in a real-world robotics application. Hence adding noise to the original maze doesn't directly capture the property we're testing (the lunar rover task addresses the other kinds of noise encountered in real-world settings, such as inaccuracies from sensor data).
> >
> > Nevertheless, we have now rerun the experiment per your suggestion, and evaluated the methods under controlled noise.
> > To emulate the prediction noise, we add Gaussian noise to the original maze and enforce the value within $(0,1)$ by $clip(maze+\epsilon, 0, 1)$, where $\epsilon$ is a noise sampled from Gaussian distribution. The results of this are below. DT-VIN seems more robust in handling noise than Highway VIN and VIN. GPPN shows robustness in the short term setting, but is not able to solve the task when long term planning is required.
> >
> >
> >
> >
> > Table R12: The success rate for each method in $35 \times 35 $ 2D Maze navigation, with maze noise $\epsilon \sim \mathcal{N}(0,0.01)$.
> >
> >
> >
> >
> >
> > | Shortest Path Length | [1,100]            | [100,200]          | [200,300]          |
> > | -------------------- | ------------------ | ------------------ | ------------------ |
> > | VIN                  | $66.41_{\pm 7.25}$ | $0.00_{\pm 0.00}$  | $0.00_{\pm 0.00}$  |
> > | GPPN                 | $91.71_{\pm 1.33}$ | $0.21_{\pm 0.27}$  | $0.00_{\pm 0.00}$  |
> > | Highway VIN          | $86.67_{\pm 4.92}$ | $57.50_{\pm 6.59}$ | $46.40_{\pm 11.2}$ |
> > | DT-VIN (ours)        | $99.21_{\pm 0.00}$ | $98.17_{\pm 0.01}$ | $97.77_{\pm 0.23}$ |
> >
> >
> >
> >
> >
> >
> > Table R13: The success rate for each method in $35 \times 35 $ 2D Maze navigation, with maze noise $\epsilon \sim \mathcal{N}(0,0.04)$.
> >
> >
> > | Shortest Path Length | [1,100]            | [100,200]           | [200,300]           |
> > | -------------------- | ------------------ | ------------------- | ------------------- |
> > | VIN                  | $59.11_{\pm 7.94}$ | $0.00_{\pm 0.00}$   | $0.00_{\pm 0.00}$   |
> > | GPPN                 | $87.43_{\pm 1.34}$ | $0.15_{\pm 0.48}$   | $0.00_{\pm 0.00}$   |
> > | Highway VIN          | $81.76_{\pm 4.74}$ | $49.5_{\pm 6.05}$   | $35.7_{\pm 11.3}$   |
> > | DT-VIN (ours)        | $95.17_{\pm 0.00}$ | $92.29_{\pm 0.57}$  | $91.99_{\pm 0.27}$  |

---

> > ### Author Response · Authors · 2024-11-29
> > **Further Response to the Reviewer Pndq (2/3)**
> >
> > ---
> >  ### Q6:
> > > ... This seems like a fairly comprehensive answer to the concern about the estimation for l being a potential weakness in the method. Are similarly strong results seen in the other environments, eg continuous control? (Not asking for new experiments to be run or added to the paper, just curious if you'd tried this and had any anecdotal evidence?)
> >
> > ### Answer:
> > Thank you for pointing this out! We were also curious about whether the strong results on the estimation of l apply to the continuous control experiments as well. In addition to the previous results, we report in Table R14 that the performance of DT-VIN loses $8%$ of success rate when l is set to 0. While not quite as promising as the other setting, this is still a strong result. We will add this to the camera ready.
> >
> > Table R14: The performance of each method on continuous control tasks.
> >
> >
> >
> > |                                 | Final Distance to the Goal (Euclidean Distance) | Success Rate (%)           |
> > | :-----------------------------: | :---------------------------------------------: | --------------------------- |
> > |               VIN               |               ${5.12_{\pm 3.19}}$               | ${62.00_{\pm 2.18}}$        |
> > |              GPPN               |               ${4.12_{\pm 2.18}}$               | ${68.12_{\pm 4.17}}$        |
> > |           Higwhay VIN           |                $4.98_{\pm 3.28}$                | ${67.31_{\pm 3.28}}$        |
> > | DT-VIN (ours) ($\widehat{l}=0$) |            $\mathbf{3.17 \pm 2.29 }$            | $\mathbf{74.00_{\pm 4.79}}$ |
> > | DT-VIN (ours) ($\widehat{l}=l$) |            $\mathbf{2.28 \pm 1.2 }$             | $\mathbf{82.00_{\pm 3.89}}$ |
> >
> >
> >
> >
> >
> > ---
> >  ### Q7:
> > > Small question on Table R7 for my own understanding - where the baselines require, eg, 710GB of GPU memory, how is this achieved on a single A100 (which only has 40 or 80GB VRAM)?
> >
> > ### Answer:
> >
> > This is a typo. We thank you for pointing it out and we have now fixed it in the paper. The baseline, GPPN, requires 710GB of GPU memory and uses 10 A100 GPUs. The updated tables below show the GPU memory consumption and training time (Wall-Clock and GPU Hours) with NVIDIA A100 GPUs for DT-VIN and the baselines. The total GPU hours for our method are comparable to those of GPPN and Highway VIN.
> >
> > Table R15: computational complexity during traning of each method, employing 600 layers and trained over 30 epochs, evaluated in a $35 \times 35$ 2D maze navigation.
> >
> >
> >
> >
> > | Method      | GPU Memory (GB) | Wall-Clock Training Time (h) | GPU Hours (h) |
> > |-------------|-----------------|----------------------------------|---------------|
> > | VIN         | 4.2             | 8.4                              | 8.4           |
> > | GPPN        | 182             | 4.2                              | 12.6          |
> > | Highway VIN | 41.3            | 14.3                             | 14.3          |
> > | DT-VIN      | 53.3            | 12.1                             | 12.1          |
> >
> >
> >
> > Table R16: Computational complexity during training of each method using 5000 layers and training for 90 epochs, evaluated on a $100 \times 100$ 2D maze navigation.
> >
> > | Method        | GPU Memory (GB) | Wall-Clock Time (h) | GPU Hours (h) |
> > |---------------|-----------------|-------------------------|---------------|
> > | VIN           | 35              | 36                      | 36            |
> > | GPPN          | 710             | 31                      | 310           |
> > | Highway VIN   | 111             | 112                     | 224           |
> > | DT-VIN (ours) | 182             | 98                      | 294           |

---

> > ### Author Response · Authors · 2024-11-29
> > **Further Response to the Reviewer Pndq (3/3)**
> >
> > ---
> >  ### Q8:
> > >  ...This doesn't fully answer the question of how the dataset scales with the problem complexity, though. Is this also linear (or close to linear?) (I realise it's probably not possible to answer this empirically in the given timeframe, am happy for this to remain unanswered - unless there is related evidence in other papers that can be cited as an indication?
> >
> >
> >
> >
> >
> >
> >
> >
> > ### Answer:
> >
> > This is a very good point. To address this, we've rerun our experiments on the mazes of size $15 \times 15$, $25 \times 25$, and $35 \times 35$, on increasingly larger numbers of expert steps (a step here is one transition from a cell to another cell).
> > Specifically, for each maze size, we look for the smallest $n$ such that, with $5000 \times n$ different mazes (with each maze having a number of expert trajectories and larger mazes having more expert trajectories), DT-VIN can achieve a $\geq 98%$ success rate on the longest length planning problem for each maze. The results of this experiment are below (and we will add this to a camera-ready version of the paper). While this is a bit of a coarse result, comparing the length column to the expert data column, we see a growth rate that looks a more than linear, but still indicates a manageable degree of complexity increase. We would like to note that using more mazes in the training set with fewer expert trajectories for each maze might further increase the sample efficiency of our method and the baselines. Designing an appropriate curriculum for this could be an exciting area of future work.
> >
> >
> >
> >
> >
> >
> >
> > Table R17: Minimal required size of dataset for DT-VIN across various maze sizes.
> >
> >
> >
> >
> >
> >
> > | Maze Size | Longest Length of Optimal Path | Required Number of Mazes | Required Amoust of Expert Steps |
> > | :-------: | :----------------------------: | :----------------------: | :----------------------------: |
> > |    15     |               80               |           15K            |               4M               |
> > |    25     |              200               |           15K            |              24M               |
> > |    35     |              300               |           10K            |              45M               |
> >
> >
> >
> >
> >
> > ---
> >  ### Q9:
> > > There's a small typo in the description here: (table 14 in the paper) "Table R9: The success rate of DT-VIN across various dataset depths, maintaining the same size as the original dataset." - should presumably read "model depths".
> >
> > ### Answer:
> > Thank you for pointing this out. We've now corrected it to "model depths" and updated the paper accordingly.
> >
> >
> > ---
> >  ### Q10:
> > > (on the different transition kernels)...
> > Am I right that DT-VIN (and others) score highest on Diff Drive, then NEWS, then Moore? Does this indicate an inverse relationship between size of action space and performance? (Asking mostly for my own interest.)
> >
> > ### Answer:
> >
> >
> >
> > These results are not directly comparable because it's not abundantly clear how to compare one kernel to one another due to the different optimal path lengths they induce (whose relative magnitude will depend on the specific configuration of the walls).
> > So it's hard to say much about Diff Drive vs. NEWS, but it is clear that here MOORE is worse than either.
> > Thus your hypothesis may hold and would be consistent with the increased complexity a larger action space brings to a task, but it would hard to confirm without a large dedicated study, which is beyond the scope of this work.
> >
> >
> >
> >
> >
> >
> >
> >
> >
> >
> >
> >
> > ---
> >
> >
> > Thank you again for your very thorough responses and engagement. We believe that addressing all your points has allowed us to make this work much, much stronger.

---

> > > ### Comment · Reviewer_Pndq · 2024-12-03
> > >
> > > Thank you for all these extra clarifications and results - I appreciate the work that has gone into this, and the time you've spent answering my questions. I have no further concerns, good luck with the paper!

---

### Official Review · Reviewer_pQwz · 2024-11-01

**Soundness:** 2
**Presentation:** 3
**Contribution:** 3
**Rating:** 5
**Confidence:** 4

**Summary:**

Summary: This paper presents the Dynamic Transition Value Iteration Network (DT-VIN), an improvement on the Value Iteration Network (VIN) designed to perform planning-based reasoning in reinforcement learning. The proposed method enhances VIN to handle larger-scale, long-term planning tasks by introducing a dynamic transition kernel (a learnable transition mapping module) and adaptive highway loss, a loss structure that connects hidden layers directly to the final loss. The effectiveness of DT-VIN is demonstrated empirically in several tasks, including 2D maze navigation, 3D VizDoom, and rover navigation.

**Strengths:**

Strengths:

1. The results on scaling up to 5000 layers are promising, showing DT-VIN's capability to solve larger-scale environments without significant performance drops, even as trajectory length increases.

2. The paper introduces simple yet innovative modifications that effectively improve VIN.

3. Even with the scale-up, DT-VIN still has model size advantage to other RL planning algorithms

4. The writing is generally clear, with well-organized ideas that communicate the methodology and results effectively.

**Weaknesses:**

1. **Weaknesses:**
   One of the key aspects of the paper is the use of the adaptive highway network and the adaptive highway loss. However, more structured experiments on the structure will be beneficial to justify the new method. Below are some details about what specifically you can do:

   1. Additional ablation studies would clarify the benefits of directly connecting hidden layers to the final loss instead of intermediate layers. Furthermore, the claim that having \( n > l \) is advantageous for short-term planning tasks needs stronger empirical backing. It would be beneficial to include more controlled experiments to justify this claim. It would strengthen the claim if there is a performance gap between the two methods in 35x35 and 100x100 2D maze environments in the “shorter SPL” regime. i.e., if you do not have \( n > l \) in your loss, does the performance drop in the [1,100] regime for 35x35 and [1,600] regime for 100x100? To be clear, I am asking for the 2D maze environments to be tested under these different regimes with success rate as the metric as in the paper.

   2. The adaptive highway network depends on an estimation of \( l \) (the trajectory/planning path length). It would be insightful to include robustness analysis of the method against different \( l \) values, exploring how sensitive DT-VIN’s performance is to the estimation of \( l \). For example, how does the success rate change in Vizdoom when you overestimate/underestimate \( l \), or when the estimate of \( l \) has high variance? The metric is the success rate.

   3. Although adaptive highway loss was introduced to address gradient stability, the ablation study suggests that the softmax latent transition kernel plays a larger role in mitigating gradient issues. Some theoretical justifications or more explanation of this gap would be helpful. Including loss curves/gradients norm graphs on one or more of the environments from the paper would help visually illustrate the trend of gradient instability for lack of each ingredient. The specific environment should be selected by the author and I would be happy with anything from the paper.

2. Testing on additional environments, such as continuous control domains or graph-based planning tasks, would provide a more comprehensive evaluation of DT-VIN. The original VIN paper included both types - learning policies with guided policy search with unknown dynamics and simulating in Mujoco for the continuous control domain (metric is measuring the final distance to the target); WebNav for the graph-based planning task (metric is success rates of the learned policy with DT-VIN for a query).

**Questions:**

None

---

> ### Author Response · Authors · 2024-11-24
> **Response to Reviewer pQwz (1/3)**
>
> We sincerely thank reviewer pQwz for the valuable comments.
> Below we address the questions and concerns.
>
>
> ---
>  ### W1.1:
> > Additional ablation studies would clarify the benefits of directly connecting hidden layers to the final loss instead of intermediate layers. Furthermore, the claim that having ( $n > l$ ) is advantageous for short-term planning tasks needs stronger empirical backing. It would be beneficial to include more controlled experiments to justify this claim. It would strengthen the claim if there is a performance gap between the two methods in 35x35 and 100x100 2D maze environments in the “shorter SPL” regime. i.e., if you do not have ( $n > l$ ) in your loss, does the performance drop in the [1,100] regime for 35x35 and [1,600] regime for 100x100? To be clear, I am asking for the 2D maze environments to be tested under these different regimes with success rate as the metric as in the paper.
>
> ### Answer:
>
> Following your suggestion, we conducted an ablation study for adaptive highway loss by evaluating the following variants in shorter planning tasks:
>
> 1. Implement skip connections for intermediate layers of the planning module of VIN, like what has been done in Residual Nets [r1] and Highway Nets [r2].
> As shown in the table below, this variant performs poorly, achieving only 61.35% success rate in comparison to DT-VIN's 99.98% on $100 \times 100$ Maze.
> These results are consistent with those in existing work [r3].
>
> 2. $\mathbf{1}_{ \lbrace n = l \rbrace }$, only building highway loss for a specific layer $n$ which satisfies $n=l$.
> As shown in the table below, this variant performs  worse than the adaptive highway loss, showing that the component $n>l$ plays an important role in the performance.
>
> 3. Building highway loss for all intermediate layers $n$, without the term $ \mathbf{1}_{ \lbrace n \ge l \rbrace }$. This variant is already verified to be less effective in the original paper, which is called "Fully Highway Loss" (see Figure 4 (c) or Section 4.1, Ablation Study, Adaptive Highway Loss).
>
>
>
>
>
> Table R1: The success rates for the variants of adaptive highway loss.
>
>
>
>
> |                                                 | $35\times 35$ 2D Maze with SPL range [1,100] | $100\times 100$ 2D Maze with SPL range [1,600] |
> | ----------------------------------------------- | :---------------------------------------: | :-----------------------------------------: |
> | Skip Connections for intermediate layers    |            $90.35_{\pm 2.53}$             |             $61.35_{\pm 3.43}$              |
> | ${\mathbf{1}}_{ \lbrace n \geq l \rbrace }$ (Adaptive Highway Loss)      |        $\mathbf{100.00_{\pm0.00}}$        |         $\mathbf{99.98_{\pm 0.00}}$         |
> | $1_{ n=l   }$ (without ${{1}}_{ \lbrace n \geq l \rbrace }$)                  |            $98.35_{\pm 2.23}$             |             $92.81_{\pm 3.78}$              |
> | Without ${\mathbf{1}}_{ \lbrace n \geq l \rbrace }$ (Fully Highway Loss) |             $98.11_{\pm1.23}$             |              $91.11_{\pm2.00}$              |
>
>
>
>
> [r1] He K, et al. Deep residual learning for image recognition[C]. CVPR. 2016.
>
> [r2] Srivastava R K, et al. Training very deep networks[C]. NeurIPS, 2015.
>
> [r3] Wang Y, et al. Highway Value Iteration Networks[C]. ICML, 2024.

---

> ### Author Response · Authors · 2024-11-24
> **Response to Reviewer pQwz (2/3)**
>
> ---
>  ### W1.2:
> > The adaptive highway network depends on an estimation of ($l$) (the trajectory/planning path length). It would be insightful to include robustness analysis of the method against different ($l$) values, exploring how sensitive DT-VIN’s performance is to the estimation of ($l$). For example, how does the success rate change in Vizdoom when you overestimate/underestimate ($l$), or when the estimate of ($l$) has high variance? The metric is the success rate.
>
> ### Answer:
>
>
> Thanks for this insightful comment.
> Note that the knowledge of the length $l$ of the expert path naturally exists in the imitation learning case.
> However, for the case where such information is unknown, one can use either the length of non-expert data or some heuristic methods to estimate $l$ when the actual $l$ is completely unknown, e.g., using the distance between the start and the goal position.
> To further address your comment, we have now run a number of ablations on this.
>
>
>
>
> First, to measure the effect of overestimation/underestimation, we conducted the experiment with various estimated values of the length of the shortest path $\widehat{l}$, which are $0, l/2, l, 2l, N$ (where $l$ is the actual length of the shortest path, $N$ is the depth of the planning module).
> Second, to evaluate the case when the etsimation of $l$ has variance, we use $l \cdot \max(\epsilon, 0)$ as the estimation, with $\epsilon$ sampled from a Gaussian distribution $\mathcal{N}(1, 1)$.
> Third, we also assessed two additional variants for estimating $l$:
> (a) One variant utilizes the length of non-expert trajectories for $l$;
> (b) Another variant estimates the shortest path length heuristically using the L1 distance between the start  $(x_s,y_s)$ and the goal $(x_g, y_g)$, i.e., $D=|x_s-x_g|+|y_s-y_g|$.
>
>
> As indicated in the table below, both overestimation and underestimation led to a performance degradation of no more than 7%. Additionally, we found that leveraging non-expert data or the heuristic L1 distance only yields a nearly 3% degradation in performance, and performs better than the case when the optimal length is extremely overestimated/underestimated. These results imply that employing the information from non-expert data or heuristic estimation could be taken as an alternative when the optimal length is not available.
>
>
> Table Rx: The ablation study for using various estimated lengths of optimal paths for adaptive highway loss, under $35 \times 35 $ ViZDoom navigation.
> The best results are highlighted by **bold**, while the second-best result is highlighted by $\underline{\text{underline}}$.
>
>
>
>
>
>
>
> | Shortest Path Length                                         | [1,100]                     | [100, 200]                  | [200,300]                   |
> | ------------------------------------------------------------ | --------------------------- | --------------------------- | --------------------------- |
> | $\widehat{l}=0$ (connected to all hidden layers)             | $99.49_{\pm 0.35}$          | $94.51_{\pm 0.77}$          | $89.1_{\pm 3.56}$           |
> | $\widehat{l}=l/2$                                            | $99.62_{\pm 0.91}$          | $96.21_{\pm 0.44}$          | $91.24_{\pm 1.68}$          |
> | ${\widehat{l}=l}$                                            | $\mathbf{99.67_{\pm 0.22}}$  | $\mathbf{97.92_{\pm 0.11}}$ | $\mathbf{96.41_{\pm 0.37}}$ |
> | $\widehat{l}=2*l$                                            | $99.61_{\pm 0.18}$          | $96.29_{\pm 0.48}$          | $93.12_{\pm 0.73}$          |
> | $\widehat{l}=N$ (connected to only last layer)               | $99.52_{\pm 0.29}$          | $95.52_{\pm 0.86}$          | $91.12_{\pm 1.64}$          |
> | $\widehat{l}=l \cdot \max(\epsilon,0), \epsilon \sim \mathcal{N}(1,1)$ | $99.62_{\pm 0.50}$          | $96.19_{\pm 0.15}$          | $93.21_{\pm 0.92}$          |
> | $\widehat{l}=len($non-expert path$)$                         | ${99.62_{\pm 0.12}}$        | $\underline{97.01_{\pm 0.69}}$        | ${93.31_{\pm 0.31}}$        |
> | $\widehat{l}=D$ (L1 distance)                            | $\underline{99.64_{\pm 0.49}}$ | ${96.92_{\pm 0.05}}$ | $\underline{93.52_{\pm 0.87}}$ |

---

> ### Author Response · Authors · 2024-11-24
> **Response to Reviewer pQwz (3/3)**
>
> ---
>  ### W1.3:
> > Although adaptive highway loss was introduced to address gradient stability, the ablation study suggests that the softmax latent transition kernel plays a larger role in mitigating gradient issues. Some theoretical justifications or more explanation of this gap would be helpful. Including loss curves/gradients norm graphs on one or more of the environments from the paper would help visually illustrate the trend of gradient instability for lack of each ingredient. ...
>
> ### Answer:
>
> Thanks for this insightful comments.
> The softmax operation ensures that the values of the dynamic transition kernels remain within $[0,1]$, helping to prevent the gradient exploding problem.
> In our experiments, we found that the gradient of DT-VIN lacking softmax operation explodes at the first forward-backward pass of training, resulting in the loss escalating to NaN (Not a Number) during the training process.
> Tables R2 and R3 shows the gradient and the loss of with and without Softmax Operation.
>
> The adaptive highway loss improves gradient flow toward shallower layers. As shown in Tables R4 and R5, without adaptive highway loss, the L1 norm of the gradient for DT-VIN is closer to zero in the first 10 layers of the network. The adaptive highway loss can reduce this vanishing gradient problem, resulting in lower loss.
>
> We have also presented the figures of these results in Section C.4, Figures 12 and 13 in the revised paper.
>
>
>
>
> Table R2: The *L1 norm of gradient* averaged over the first 10 layers during the training process for DT-VIN with and without **Softmax Operation**.
>
> | Epoch                     |  1   |  10  |  20  |  30  |
> | ------------------------- | :--: | :--: | :--: | :--: |
> | With Softmax Operation         | 0.97 | 0.15 | 0.11 | 0.09 |
> | Without Softmax Operation | inf  | NaN  | NaN  | NaN  |
>
>
>
>
> Table R3: The *loss* during the training process for DT-VIN with and without **Softmax Operation**, evaluated on $35 \times 35$ 2D maze with depth $N=600$.
>
> | Epoch                     |   1   |  10   |  20   |  30   |
> | ------------------------- | :---: | :---: | :---: | :---: |
> | With Softmax Operation    | 1.112 | 0.355 | 0.353 | 0.352 |
> | Without Softmax Operation |  NaN  |  NaN  |  NaN  |  NaN  |
>
>
>
>
>
>
>
>
> Table R4: The *L1 norm of gradient* averaged over the first 10 layers during the training process for DT-VIN with and without **Adaptive Highway Loss**.
>
>
>
> | Epoch                         |  1   |  10  |  20  |  30  |
> | ----------------------------- | :--: | :--: | :--: | :--: |
> | Adaptive Highway Loss         | 0.97 | 0.15 | 0.11 | 0.09 |
> | Without Adaptive Highway Loss | 0.03 | 0.013 | 0.011 | 0.008 |
>
>
>
>
>
>
> Table R5: The *loss* during the training process for DT-VIN with and without **Adaptive Highway Loss**.
>
>
> | Epoch                         |   1   |  10   |  20   |  30   |
> | ----------------------------- | :---: | :---: | :---: | :---: |
> | Adaptive Highway Loss         | 1.112 | 0.355 | 0.349 | 0.341 |
> | Without Adaptive Highway Loss | 1.111  | 0.387 | 0.367 | 0.359 |
>
>
>
>
>
>
> ---
>  ### W2:
> > Testing on additional environments, such as continuous control domains or graph-based planning tasks, would provide a more comprehensive evaluation of DT-VIN. The original VIN paper included both types - learning policies with guided policy search with unknown dynamics and simulating in Mujoco for the continuous control domain (metric is measuring the final distance to the target....
>
> ### Answer:
>
>
> In response to your suggestion for diversity in our experimental domains, we have extended our evaluation to a continuous control setting, where both state and action spaces are continuous. Here, we use the Point Maze as the benchmark [r1], where the agent must decide the forces to apply to a controllable ball to reach a goal within 800 steps. We measure both the final distance to the goal and the success rate, where we say an agent has succeeded if the Euclidean distance to the goal is less than 0.5.
> As shown in the table below, DT-VIN achieved closer proximity to the goal and a higher success rate compared to the other methods.
>
>
>
>
>
> Table R6: The performance of each method on continuous control tasks.
>
>
> |               | Final Distance to the Goal (Euclidean Distance) | Success Rate (%)           |
> | :-----------: | :----------------------------------------: | --------------------------- |
> |      VIN      |            ${5.12_{\pm 3.19}}$             | ${62.00_{\pm 2.18}}$        |
> |     GPPN      |            ${4.12_{\pm 2.18}}$             | ${68.12_{\pm 4.17}}$        |
> |  Higwhay VIN  |             $4.98_{\pm 3.28}$              | ${67.31_{\pm 3.28}}$        |
> | DT-VIN (ours) |          $\mathbf{2.28 \pm 1.2 }$          | $\mathbf{82.00_{\pm 3.89}}$ |
>
>
>
> [r1] Fu, Justin, et al. ”D4RL: Datasets for Deep Data-Driven Reinforcement Learning.” arXiv preprint arXiv:2004.07219 (2020)

---

> > ### Comment · Reviewer_pQwz · 2024-11-27
> >
> > I still have some concerns about the novelty and experimental validation. The architectural contributions (incorporating observations into transition kernel, using convolutional layers) appear to be directly derived from the original VIN paper (as pointed out by another reviewer). Additionally, the possible existence of pre-computed optimal paths via Dijkstra's for evaluation raises questions about the practical utility of the proposed method. The train-test split and reliance on other "ground truth solvers" is not entirely clear to me right now.
> >
> > I do appreciate the additional clarifications the authors provided, which did answer many of my open questions. I would say that the above are really my only outstanding issues with the current manuscript.

---

> > > ### Author Response · Authors · 2024-11-29
> > > **Further Response to the Reviewer pQwz (1/3)**
> > >
> > > We thank the Reviewer pQwz for their continued engagement. We have prepared some clarifications to help resolve your concerns below.

---

> > > ### Author Response · Authors · 2024-11-29
> > > **Further Response to the Reviewer pQwz (2/3)**
> > >
> > > ---
> > >  ### Q1:
> > > > Concerns about the novelty (observations-depdent, CNN).
> > >
> > > ### Answer to Q1 (1/2):
> > >
> > >
> > > As we've also now pointed out to reviewer FzU7, we think there has been a bit of confusion here and we apologize for not being more clear in our earlier comments.
> > > Our proposed "*dynamic transition kernel*" goes well beyond the the "*observation-dependent transition kernels*" that are partly described in VIN.
> > > While observation-dependent transition kernels and the dynamic transition kernels we propose here both depend on the observation $x$ (formally, $\overline{\mathsf{T}}=f^{\overline{\mathsf{T}}}(x)$),
> > > **the main innovation of our approach lies in the fact that our dynamic kernels are also a function of the latent state.
> > > In contrast, the observation-dependent kernel of VIN are not.**
> > >
> > > Formally, this means that (as shown in eq. (1) of the paper and reproduced below), the latent transition kernel ${\overline{\mathsf{T}}}_{i,j,\overline{a},i^{\prime},j^{\prime}}$ is different for each latent state $(i,j)$:
> > >
> > >
> > > $$\overline{V}\_{i,j}^{(n)} = \max\_{\overline{a}} \sum\_{i',j'} \overline{\mathsf{T}}\_{i,j,\overline{a},i',j'}^{\text{dyn}}
> > > \left( \overline{\mathsf{R}}\_{i-i',j-j'} + \overline{V}\_{i-i',j-j'}^{(n-1)} \right)$$
> > >
> > >
> > >
> > >
> > > **This novel dynamic formulation for the kernel greatly improves the representational capacity of the kernel and is never proposed or implemented in the original VIN nor---to the best of our knowledge---in any later studies.**
> > > In VIN, the transition kernel remains invariant across different latent states, meaning the same transition kernel ${\overline{\mathsf{T}}}_{\overline{a},i^{\prime},j^{\prime}}$ is used for every different latent state $(i,j)$.
> > >
> > > [
> > > Note that in the [VIN paper](https://proceedings.neurips.cc/paper/2016/file/c21002f464c5fc5bee3b98ced83963b8-Paper.pdf), they use $W_{l, i, j}^{\overline{a}}$ is used instead of ${\overline{\mathsf{T}}}_{\overline{a},i^{\prime},j^{\prime}}$; see the second paragraph of Sec. 3.1 in the original the [VIN paper](https://proceedings.neurips.cc/paper/2016/file/c21002f464c5fc5bee3b98ced83963b8-Paper.pdf).
> > > ]
> > >
> > >
> > > We summarize this again below.
> > > The transition kernels can be additionally dependent on **(1) the observation**, and **(2) the latent state**.
> > > In VIN, they propose transition kernels that are dependent on **(1) the observation** but they then formalize kernels that are independent of **(1) the observation**.
> > > All their kernels are independent of **(2) the latent state**.
> > > We propose kernels that are dependent on both **(1) the observation**, and **(2) the latent state**.
> > > The key innovation here is the dynamic nature of our kernels that comes from **(2) the latent state**.
> > > The below table summarizes the differences between the kernels.
> > >
> > >
> > >
> > >
> > > Table R7: A comparison of different latent transition kernels.
> > >
> > >
> > >
> > > | Types of Kernels                  | Comments                                   | dependent on observation? | dependent on the latent state? |
> > > | --------------------------------- | ------------------------------------------ | :-------------------: | :---------------------: |
> > > | invariant transition kernel       | proposed and implemented in VIN                         |          NO           |           NO            |
> > > | observation-dependent kernel      | partly proposed in VIN paper, but not implemented |          **YES**          |           NO            |
> > > | latent state-dependent kernel      | used here for ablation study |          NO          |           **YES**            |
> > > | dynamic transition kernel  (ours) | proposed and implemented in our paper                      |          **YES**          |           **YES**           |
> > >
> > >
> > >
> > > To further demonstrate the importance of this difference, we've run an ablation study to compare the four variants (i.e., independent of (1) and (2), dependent on (1) but not (2), dependent on (2) but not (1), and dependent on both (1) and (2)). The results are shown below. Here the importance of this surgical difference is apparent, with removing dependency on either (1) or (2) leading to a rapid collapse.
> > >
> > >
> > >
> > > Table R8: The success rate for each method in 35×35 2D maze navigation.
> > >
> > >
> > > | Shortest Path Length              | [1,150]                      | [150,300]                    | [300,500]                   |
> > > | --------------------------------- | ---------------------------- | ---------------------------- | --------------------------- |
> > > | invariant transition kernel       | $68.44_{\pm 3.12}$           | $0.03_{\pm 0.01}$            | $0.00_{\pm 0.00}$           |
> > > | observation-dependent kernel      | $71.44_{\pm 4.31}$           | $0.09_{\pm 0.02}$            | $0.00_{\pm 0.00}$           |
> > > | latent state-dependent kernel     | $73.44_{\pm 3.09}$           | $1.44_{\pm 0.37}$            | $0.00_{\pm 0.00}$           |
> > > | dynamic transiiton kernel  (ours) | $\mathbf{100.00_{\pm 0.00}}$ | $\mathbf{99.99_{\pm 0.01}}$ | $\mathbf{99.77_{\pm 0.23}}$ |

---

> > > ### Author Response · Authors · 2024-11-29
> > > **Further Response to the Reviewer pQwz (3/3)**
> > >
> > > ### Answer to Q1 (2/2):
> > >
> > >
> > >
> > > Regarding the convolution layers, we believe that this is also a minor misunderstanding around this statement we made:
> > >
> > > > Moreover, the original VIN paper did not specify whether one should employ fully connected or convolutional layers for the transition mapping.
> > >
> > > By "convolutional layers for the transition mapping" we mean that the architecture of the transition mapping module utilizes a CNN architecture. Here, we are not talking about the architecture of the *VI module (planning module)* of VIN, which also uses a CNN architecture (in our paper and in the original VIN paper).
> > >
> > > Although the original VIN proposes a general framework for using an observation-dependent transition kernel in their Method section (Section 3), they did not define the architecture of the transition mapping module---that is, whether it should use fully-connected layers or CNN layers.
> > > This is because, in their experiments, the transition kernel is not only invariant with respect to each latent state (as previously mentioned) but also does not depend on the observation (refer to footnote 3 in Section 4 of the original VIN paper); their transition kernel consists just of learnable parameters.
> > >
> > > Since we are the first to implement an observation-dependent transition kernel, we needed to specify the architecture of such a mapping function.
> > > This is just a minor contribution.
> > > Our major contributions are the dynamic transition kernel and using a highway loss.
> > >
> > >
> > >
> > >
> > > ---
> > >  ### Q2:
> > > > Additionally, the possible existence of pre-computed optimal paths via Dijkstra's for evaluation raises questions about the practical utility of the proposed method. The train-test split and reliance on other "ground truth solvers" is not entirely clear to me right now.
> > >
> > > ### Answer:
> > > We ensure that no mazes are shared by the training and test datasets.
> > > Specifically, while each unique arrangement of walls may have various starting positions and goals, we ensure that any given wall arrangement appears exclusively in either the training set or the test set.
> > > This approach prevents any information leakage from the training set to the test set.
> > > During testing, we do not use any ground-truth data when running either our method or the baselines (e.g., we don't look at the elevation data or any expert trajectories, which might not be available or could be (like elevation data) expensive to produce); instead, we explicitly evaluate their generalization capabilities in environments wholly not encountered during training.
> > >
> > > At training time, we use Dijkstra's algorithm to produce some expert trajectories (which may not be available at test time).
> > > In other settings, expert trajectories may also just be generally available or produced with a different method like Monte Carlo etc.
> > > At test time Dijkstra's algorithm is used solely to calculate the Optimality Rate of the algorithms for external evaluation purposes (i.e., only to draw the figures in the paper).
> > > Dijkstra's algorithm only works here because we give it access to "illegal data" (elevation data/true ViZDoom 2D map) that would not be available at test time in upstream tasks.
> > > Calculating the optimality rate is important for assessing the general planning capabilities of various methods. This practice is analogous to using labeled data in a test set to evaluate a predictive model in supervised learning.
> > > Note that we would not need to use Dijkstra's at test time if we were reporting only the results in terms of Success Rate for the methods.
> > >
> > >
> > > ---
> > >  ### Q3:
> > > > I do appreciate the additional clarifications the authors provided, which did answer many of my open questions. I would say that the above are really my only outstanding issues with the current manuscript.
> > >
> > > ### Answer:
> > > We hope that the above has cleared everything up. We of course, remain at your disposal to clarify the above or answer any further questions you may have.
> > >
> > >
> > > We would like to thank you again for your time and effort spent reviewing our paper.

---

> ### Author Response · Authors · 2024-12-02
> **Discussion Period Ending Soon**
>
> Dear reviewer pQwz, as the discussion period is nearing its end, please let us know if our recent rebuttal has successfully put to rest your concerns. We have made extensive efforts to address both your original and new comments, revising the paper as needed (in accordance with the expectations of the ICLR reviewing process). We understand you likely have many pressing responsibilities, and we appreciate the time you've already dedicated to reviewing our work. We are ready to respond if there are any remaining questions. Thank you once again for your valuable feedback.

---

### Official Review · Reviewer_vguh · 2024-11-04

**Soundness:** 3
**Presentation:** 3
**Contribution:** 3
**Rating:** 6
**Confidence:** 3

**Summary:**

This paper identifies the reasons for VIN's deficiency in long-term and large-scale planning tasks as (1) the representation capacity of the latent MDP and (2) the planning module's depth. The paper mitigates the issues with a dynamic transition kernel and adaptive highway loss. As a result, the paper scales VIN to 5000 layers and solves large-scale planning tasks.

**Strengths:**

The paper effectively identifies a significant limitation of the VIN method: its difficulty in scaling to long-term and large-scale planning tasks. While the proposed methods are straightforward, they are well-motivated and demonstrate substantial effectiveness in addressing this limitation. The experimental results support the claims, showing clear improvements in both scalability and performance. This combination of simplicity and efficacy highlights the practicality of the approach.

**Weaknesses:**

- Clarification of Motivation: While the motivations for introducing a dynamic transition kernel and an adaptive highway loss are clearly explained, the significance of investigating the Value Iteration Network (VIN) remains unclear. In the introduction, the paper notes that "Within reinforcement learning, one notable method is the Value Iteration Network (VIN)," but the paper does not expand on why VIN is particularly notable or important to study, especially in comparison to other established methods like Dreamer and MuZero. Adding a discussion that situates VIN’s relevance among these methods could strengthen the motivation and context for readers.
- Clarity of Equation 2: It would be helpful to present the overall loss function in addition to the adaptive highway loss, as showing only $\mathcal{L}$ here may introduce ambiguity. Clarifying the meaning of x and y in Equation 2 is also necessary.
- Transition from RL to Imitation Learning: The paper begins with a focus on reinforcement learning in the Introduction and Preliminaries sections, but it transitions to imitation learning (IL) in Section 3.2 and in the experimental setup without sufficient explanation. This shift from RL to IL feels abrupt and could be confusing. A more explicit discussion is needed to bridge RL and IL.

**Questions:**

See weaknesses above.

---

> ### Author Response · Authors · 2024-11-24
> **Response to Reviewer vguh**
>
> We appreciate the insightful and helpful comments from Reviewer vguh.
> Please find our responses to your concerns and questions below.
>
>
> ## Weaknesses
>
> ---
>  ### W1:
> > Clarification of Motivation: While the motivations for introducing a dynamic transition kernel and an adaptive highway loss are clearly explained, the significance of investigating the Value Iteration Network (VIN) remains unclear. In the introduction, the paper notes that "Within reinforcement learning, one notable method is the Value Iteration Network (VIN)," but the paper does not expand on why VIN is particularly notable or important to study, especially in comparison to other established methods like Dreamer and MuZero. Adding a discussion that situates VIN’s relevance among these methods could strengthen the motivation and context for readers.
>
> ### Answer:
>
>
> Thank you for the constructive feedback.
> We have revised the introduction in Section 1 to better articulate the unique position and advantages of VIN within the spectrum of AI planning techniques.
> VIN stands out in the reinforcement learning landscape due to its unique architecture, which integrates the latent planning module into the deep neural network.
> Unlike Dreamer, which leverages learned models to simulate or predict future states [r1], and MuZero, optimizes policies through a sophisticated tree-based search strategy using predictions from its learned model [r2]. VIN's integrated architecture allows it to be trained end-to-end, meaning that both the state representation components and the planning components are trained simultaneously. Moreover, the integrated planning capability of VINs enables them to effectively generalize to related but unseen tasks.
>
> [r1] Hafner, Danijar, et al. "Dream to control: Learning behaviors by latent imagination." ICLR, 2020.
>
> [r2] Schrittwieser, Julian, et al. "Mastering atari, go, chess and shogi by planning with a learned model." Nature, 2020.
>
>
>
>
>
> ---
>  ### W2:
> > Clarity of Equation 2: It would be helpful to present the overall loss function in addition to the adaptive highway loss, as showing only $\mathcal{L}$ here may introduce ambiguity. Clarifying the meaning of x and y in Equation 2 is also necessary.
>
> ### Answer:
> We thank the reviewer for the suggestion and have updated the paper accordingly.
> Eq. 2 aims to provide a general form of $\mathcal{L}$ with arbitrary single-term loss function $\ell$, which depends on the task.
> For example, in imitation learning with discrete action space, where $x$ is the observation, the label $y$ is the expert action, and $\ell$ is the cross-entropy loss, the overall loss function can be rewritten as
>
> $$
> \mathcal{L}(\theta) = \frac{1}{K |\mathcal{D}|} \sum_{(x,y,l) \in \mathcal{D}} \sum_{n=1}^{N}
> 1_{\lbrace n \geq l \rbrace} \left(
> -\sum_{a \in \mathcal{A}} 1_{\lbrace a = y \rbrace} \log f^\pi \left(\overline{V}^{(n)}(x), a \right)
> \right).
> $$
>
>
> where $ f^{\pi} \left( \overline{V}^{(n)}(x), a \right) $ represents the probability of taking action $a$ given observation $x$.
>
> For reinforcement learning where the policies are learned from reward feedback, the overall loss function can be rewritten as
> $$
> \mathcal{L}\left( \theta \right) =\frac{1}{K |\mathcal{D}| }\sum_{\left( x,y,R,l \right) \in \mathcal{D}}^{}{\sum_{n=1}^{N} {
> \mathbf{1}_{ \lbrace n \geq l \rbrace }
> \Big(
>        - R \log f^{\pi} \left( \overline{V}^{(n)}(x), y \right)
>  \Big),
> }}$$
> where $y$ is the excuted action, and $R$ the cumulative future reward.
>
> The definition of $x,y$ is presented in Lines 123 and 139-142 in the *original* paper.
> We have polished our description with a clear definition and presentation in Sec 2, Sec 3.2 in the *revised* paper.
>
>
>
>
>
>
> ---
>  ### W3:
> > Transition from RL to Imitation Learning: The paper begins with a focus on reinforcement learning in the Introduction and Preliminaries sections, but it transitions to imitation learning (IL) in Section 3.2 and in the experimental setup without sufficient explanation. This shift from RL to IL feels abrupt and could be confusing. A more explicit discussion is needed to bridge RL and IL.
>
> ### Answer:
>
>
>
> Thank you for your insightful feedback. We acknowledge the oversight in transitioning from RL to IL and have addressed this by introducing IL in Section 1 and 2 to bridge this gap.
> IL, while a separate paradigm, addresses the same decision-making problems as RL. It improves our evaluation by reducing the variability caused by RL’s exploratory processes. By focusing on IL, we leverage expert demonstrations to evaluate our network’s planning capabilities in a more stable and controlled setting.

---

> > ### Comment · Reviewer_vguh · 2024-11-25
> > **Thanks for the reply**
> >
> > Thanks for the detailed reply. I have no further questions. I will keep my evaluation.

---

> > > ### Author Response · Authors · 2024-11-29
> > > **Thank you for the positive feedback**
> > >
> > > Thank you for your positive feedback. Please feel free to reach out if you have any future questions.

---

### Author Response · Authors · 2024-12-04
**Summary of the Rebuttal Discussion and Paper Revision**

Dear Reviewer vguh, pQwz, Pndq, FzU7 and Area Chairs,

  We sincerely thank you for the valuable suggestions, feedback, and the time you spent on this paper. Your feedback has clearly improved the quality of our work.

  We provide a summary of the discussions and revisions made to our paper during the discussion period.

## 1. Most recently addressed concerns



- Novelty: There was some confusion between the proposed dynamic transition kernel and VIN's observation-dependent transition kernel. **As we noted in the response and our paper, our dynamic kernel is also latent state-dependent, a feature that has *never* been proposed or implemented in the original VIN nor --- to the best of our knowledge---in any later studies.** Our experimental results also show this novel dynamic formulation for the kernel greatly improves the representational capacity. See our answer in "Further Response to the Reviewer pQwz (2/3), Q1" and "Further Response to the Reviewer FzU7 (2/3), Q5" for more details.

- The difference between train and test dataset:
We clarified that the wall arrangement of the mazes appears exclusively in either the training or the test set.
See our answer in "Further Response to the Reviewer pQwz (3/3), Q2" and "Further Response to the Reviewer FzU7 (3/3), Q7".


- Concerns about using ground-truth data:
We clarified that using ground-truth data is analogous to supervised learning -- we use the labeled data to train the models in the training phase and to measure the performance of the method in the testing phase.
However, this labeled data is not required in practical applications—such as deploying the agent in an unseen 3D ViZDoom environment or analyzing a new aerial photograph of the moon—where labeled data may not be available or could be costly to obtain.
See our answer in "Further Response to the Reviewer pQwz (3/3), Q2" and "Further Response to the Reviewer FzU7 (3/3), Q8".







## 2. Previously resolved concerns


- Reviewer vguh:
    - W1: Clarified motivation. (Revised in Sec 1, Introduction)
    - W2: Clarified the loss function in eq. 2 (Revised in Sec 2, Sec 3.2)
    - W3: Clarified transiting from reinforcement learning to imitation learning (Revised in Sec 1, Sec 2)

- Reviewer pQwz:
    - W1.1: Added further ablation studies on the components in the adaptive highway loss. (Revised in Sec C.4, Ablation on Highway Loss)
    - W1.2: Conducted extensive ablation studies on the estimation of the length $l$. (Revised in Sec C.4, Ablation on the Choice of $l$)
    - W1.3: Conducted further ablation studies on the loss and gradient behaviors with and without highway loss and softmax operation. (Revised in Sec C.4, Gradient and Loss Analysis)
    - W2: Conducted experiments on continuous control. (Revised in Sec C.3)

- Reviewer Pndq:
    - W1: Clarified the paper's motivation and highlighted the differences between our method and A*.(Revised in Sec. 1)
    - W2/Q6: Experiments on continuous control. (Revised in C.3)
    - Q1:  Added additional ablation studies on the estimation of the length $l$. (Revised in Sec C.4, Ablation on the Choice of $l$)
    - Q2/Q10: Conducted additional experiments on various transition mechanisms. (Revised in Sec C.2)
    - Q3/Q7/Q8/Q9: Addressed concerns about scaling the method, including computational resources, model size, and data usage. (Revised in Sec C.5)
    - Q5: Conducted experiments with controlled noise.(We will update in the camera-ready version of the paper)




- Reviewer FzU7:
    - W2: Added additional ablation studies on the estimation of the length $l$. (Revised in Sec C.4, Ablation on the Choice of $l$)
    - W3: Concerns about the computational complexity in practice. (Revised in Sec B.1)
    - Q1: Clarified the difference between train and test dataset. (We will update in the camera-ready version of the paper)
    - Q2: Added more plots showing results for the $100 \times 100$ maze. (Revised in Fig. 9 and Fig. 10)
    - Q3: Clarified experimental details for the 3D ViZDoom setup. (Revised in Sec. 4.2 and B.2)
    - Q4: Clarified the use of elevation data on Rover Navigation. (Revised in Sec 4.3)


---
Thank you once again for the time and effort you invested in reviewing our paper. We believe your feedback has significantly strengthened it.

---

### Meta-Review · Area_Chair_5Hnt · 2024-12-24

**Metareview:**

This paper introduces DT-VIN, a method to extend value iteration networks with a dynamic transition kernel and an adaptive highway loss to tackle very long planning horizons. The empirical results show improved performance in large 2D mazes, VizDoom and rover navigation tasks.

It is interesting to see the model scale to deeper VIN modules (up to 5000 layers). The proposed method also outperforms standard VIN variants in maze like navigation and rover tasks.

The novelty is unclear as VIN originally showed observation dependent kernels. This paper does implements a latent state dependent version on top. The tasks are maze centric and they should be expanded to known environments to investigate the generality of the approach. This led to mixed scores during the review process. The method also relies on hand crafted expert paths, which is fine for testing during training but it may not have direct immediate real-world constraint applicability. I think this is a very interesting paper and presents a promising approach. I believe the paper will becomes stronger after this revision and should get better consensus for acceptance in a future conference.

**Additional Comments On Reviewer Discussion:**

Reviewers questioned novelty, data reliance, and real-world scalability. Authors provided ablations, an extra continuous-control example, and clarifications about dynamic kernels. However, the fundamental concerns about incremental novelty and maze-centric focus were only partly addressed. Hence, the final recommendation is to reject.

---

### Decision · Program_Chairs · 2025-01-22

Reject